# Darwinian Evolution of Self-Replicating DNA in a Synthetic Protocell

Zhanar Abil [1,2,6], Ana María Restrepo Sierra [1,6], Andreea R. Stan[1,7], Amélie Châne[1], Alicia del Prado [3], Miguel de Vega [3], Yannick Rondelez [4] & Christophe Danelon [1,5] ✉

Replication, heredity, and evolution are characteristic of Life. We and others have postulated that the reconstruction of a synthetic living system in the laboratory will be contingent on the development of a genetic self-replicator capable of undergoing Darwinian evolution. Although DNA-based life dominates, the in vitro reconstitution of an evolving DNA self-replicator has remained challenging. We hereby emulate in liposome compartments the principles according to which life propagates information and evolves. Using two different experimental configurations supporting intermittent or semi-continuous evolution (i.e., with or without DNA extraction, PCR, and re-encapsulation), we demonstrate sustainable replication of a linear DNA template – encoding the DNA polymerase and terminal protein from the Phi29 bacteriophage – expressed in the 'protein synthesis using recombinant elements' (PURE) system. The self-replicator can survive across multiple rounds of replication-coupled transcription-translation reactions in liposomes and, within only ten evolution rounds, accumulates mutations conferring a selection advantage. Combined data from next-generation sequencing with reverse engineering of some of the enriched mutations reveal nontrivial and context-dependent effects of the introduced mutations. The present results are foundational to build up genetic complexity in an evolving synthetic cell, as well as to study evolutionary processes in a minimal cell-free system.

A fundamental goal of modern synthetic biology is the construction of synthetic systems with life-like properties. It is also likely that the learning-by-doing approach involved in the construction of a synthetic cell will spearhead advances in biomedicine, biotechnology, and fundamental biology[1–6]. In this quest, a number of life's features have been reconstituted in a cell-free environment[7–13], although a functionally integrated, autonomous synthetic cell seems still out of reach.

One remarkable feature of extant living forms is evolution, i.e., the ability to diversify and gradually adapt to changing environments. This ability is responsible for terrestrial Life's extraordinary diversity and robustness, allowing it to colonize extremely diverse niches and survive multiple geological calamities in the past 3.5–3.8 billion years[14,15]. Importantly, it is thought that evolution, in an abiotic, molecular version also has allowed life's emergence in the first place[16–20]. We therefore asked whether in vitro evolution could be used as a tool in our efforts to build a synthetic cell and better understand living processes.

A key prerequisite for evolution is heredity. In contemporary life form, the dominant molecular mechanism supporting heredity is DNA

[1]Department of Bionanoscience, Kavli Institute of Nanoscience, Delft University of Technology, Delft, Netherlands. [2]Department of Biology, University of Florida, 882 Newell Dr, Gainesville, USA. [3]Centro de Biología Molecular Severo Ochoa (Consejo Superior de Investigaciones Científicas-Universidad Autónoma de Madrid), Nicolás Cabrera, 1, Madrid, Spain. [4]Laboratoire Gulliver, UMR7083 CNRS/ESPCI Paris-PSL Research University, 10 rue Vauquelin, Paris, France. [5]Toulouse Biotechnology Institute (TBI), Université de Toulouse, CNRS, INRAE, INSA, Toulouse, France. [6]These authors contributed equally: Zhanar Abil, Ana María Restrepo Sierra. [7]Deceased: Andreea R. Stan. ✉e-mail: danelon@insa-toulouse.fr

replication[21]. In vitro reconstitution of replication thus represents a major step in crafting synthetic living systems from the ground up[6]. However, early molecular replicators could in principle have been based on molecules other than DNA[22,23]. A variety of self-replicating non-DNA systems have been created in the laboratory. For example, non-enzymatic self-replication based on autocatalytic template production[24,25], cross-catalysing RNA replicators[26], self-replicating peptides[27–30], vesicles[31], micelles[32–34], supramolecular polymerisation[35], and cooperative replicating RNA networks[36] have been described. These studies showed that populations of molecular replicators can respond to selection pressure, exhibit exponential growth, feature emergent traits related to heredity and selection, and in a few cases undergo Darwinian evolution[26]. However, in all these systems, genotype and phenotype are manifested in the same molecule, fundamentally restricting the evolutionary potential[37].

The separation of genotype and phenotype into separate molecules is a fundamental step in the history of life, and significantly increases a system's ability to evolve[37]. A Darwinian protocell with an RNA genotype and a protein phenotype was studied in vitro by Ichihashi and colleagues, who built a translation-coupled RNA self-replication system[38]. They performed evolutionary experiments of self-replicating RNA molecules by self-encoded Qβ replicase in droplet compartments[38–43]. However, it would be challenging to develop a synthetic cell whose functions are fully encoded on an RNA-genome. Some of the reasons are: incomplete separation of genotype and phenotype (RNA folding and catalytic function), RNA is highly unstable when compared to DNA, most extant life is DNA-based, and the majority of currently available tools for regulation and processing of nucleic acids are based on systems with a DNA genome. Moreover, RNA-based Qβ-replicase system suffers from poor template generality, which can limit genome expansion to encode more functionalities.

DNA-based synthetic genome was also explored as a promising strategy for building a Darwinian protocell[7,44]. For example, transcription and translation-coupled DNA replication via rolling circle replication (RCR) has been investigated[44]. Therein, a circular DNA template encodes the bacteriophage Φ29 DNA polymerase, which gets expressed in situ to amplify its parental DNA template. However, this replication strategy produces repetitive concatemers of DNA as the replication product, thus requiring a recombination step to regenerate the original circular DNA structure for the next round of evolution[45]. Adaptive evolution of transcription and translation-coupled replicating circular DNA in the presence of a recombinase in emulsion droplets was shown to be possible[46–48]. Nevertheless, RCR coupled with recombination still results in a mixture of different DNA products with only a trace amount of monomeric circular DNA[46–48]. The dominance of (possibly non-clonal) concatemers results in a reduced enrichment efficiency[49] leading to the accumulation of inactive variants in the replicating DNA population[50,51]. Alternatively, additional DNA processing steps would be required between each round of evolution to restore the original DNA structure. Therefore, this strategy would be difficult to implement towards the evolution of larger synthetic genomes.

In contrast, some DNA viruses use relatively simple linear genome replication schemes, which enable efficient replication initiation and complete restoration of the original monomeric DNA structure after each round of replication[7,52]. Here, bacteriophage Φ29-based minimal linear DNA replication was explored as a promising strategy[7,52]. In this amplification scheme, the DNA sequence of interest is flanked by Φ29 origins at each side, and the DNA is replicated by Φ29 DNA polymerase (DNAP) in combination with Φ29 terminal protein (TP), which functions as a primer, leading to covalent protein-DNA conjugates[7,53]. With this system, amplified DNA from one round of evolution may in principle be directly carried on to the next round by re-encapsulation in fresh microcompartments as single clones. Moreover, clonal amplification may enable a more stringent genotype-phenotype link and,

thus, an enhanced variant enrichment efficiency[50]. Since reproduction of the parental DNA does not require further processing steps[7,53], it simplifies the building and evolution of long synthetic genomes. Hence, the simplicity and efficiency of using protein-primed linear DNA replication is attractive in the context of building synthetic biological systems via an evolutionary approach[5]. However, it remains unknown to what extent such a protein-primed minimal DNA self-replicator is capable of propagating over multiple generations in a cell-free environment, and supporting adaptive evolution.

Extant cells are bounded by a membrane, which provides a tight genotype-phenotype linkage while allowing controlled metabolite exchange and supporting energetic processes. These roles can be mimicked by artificial compartmentalization using liposomes[49,54–58]. In these examples, ultra-high-throughput screening of genes and their products was combined with template extraction, bulk amplification, and re-encapsulation of the template in new compartments to repeat the cycle of evolution. Although useful for in vitro directed evolution of single functional molecules, the system's level evolution of a synthetic cell would require a more streamlined approach[5]. For example, a faster, more efficient approach could be achieved with Darwinian evolution[38,46,59] (i.e., where replicators are selected based on differential self-amplification, obviating the need for high-throughput screening), and direct redistribution of self-amplified replicators into new compartments for the next cycle of evolution[38,43,45] (i.e., removing the need for template extraction and bulk amplification). Neither Darwinian evolution of replicators, nor direct redistribution of replicators in fresh compartments, to our knowledge, has been demonstrated in liposomes so far. Thus, it is not clear if in vitro Darwinian evolution of self-replicating DNA can be carried out using such a selection scheme.

In this work, we demonstrate the construction of a Darwinian synthetic protocell in the form of a liposome that mimics a living system via its ability to support the adaptive evolution of a DNA molecule expressing its own self-replication machinery. We show that the synthetic protocells can support sustained self-encoded replication and adaptive evolution via cycles of compartmentalized in vitro transcription-translation-replication (IVTTR) using a recombinant gene expression system. Moreover, to enable a more streamlined in vitro evolution procedure, we implemented a freeze-thaw cycle-based method[60] between rounds of evolution to redistribute the DNA content across vesicles. Improved self-amplification is demonstrated, providing a stepping stone for further functional integration of modules towards the construction of a self-replicating synthetic cell[5].

## Results

### Design strategy and replicator engineering

For the design of our DNA self-replicator, we drew inspiration from the replication mechanism of the *Bacillus subtilis* bacteriophage Φ29 genome. In vitro replication of heterologous DNA, flanked with Φ29 origins of replication, and in the presence of four purified Φ29 proteins has already been reported[52]. Moreover, a synthetic DNA encoding the Φ29 DNA polymerase (DNAP, from gene *p2*) and terminal protein (TP, from gene *p3*), named *ori-p2p3*, can be self-amplified when expressed in PURE system in the presence of purified auxiliary proteins (double-stranded and single-stranded binding proteins: DSB and SSB), and dNTPs[7]. The linear DNA template in this case encompasses two transcriptional units and two origins of replication, one at each end (Fig. 1a). Each gene was codon optimized for improved expression with an *E. coli*-based translation machinery and was cloned between a T7 promoter and either vsv-r1 and vsv-r2 (from vesicular stomatitis virus (VSV) internal terminator) or a T7 transcription terminator, thus constituting a chimeric, synthetic DNA construct[7].

The *ori-p2p3* template, along with PURE*frex*2.0 – the latest version of the 'protein synthesis using recombinant elements' (PURE) system – and the accessory proteins and additives for replication, were

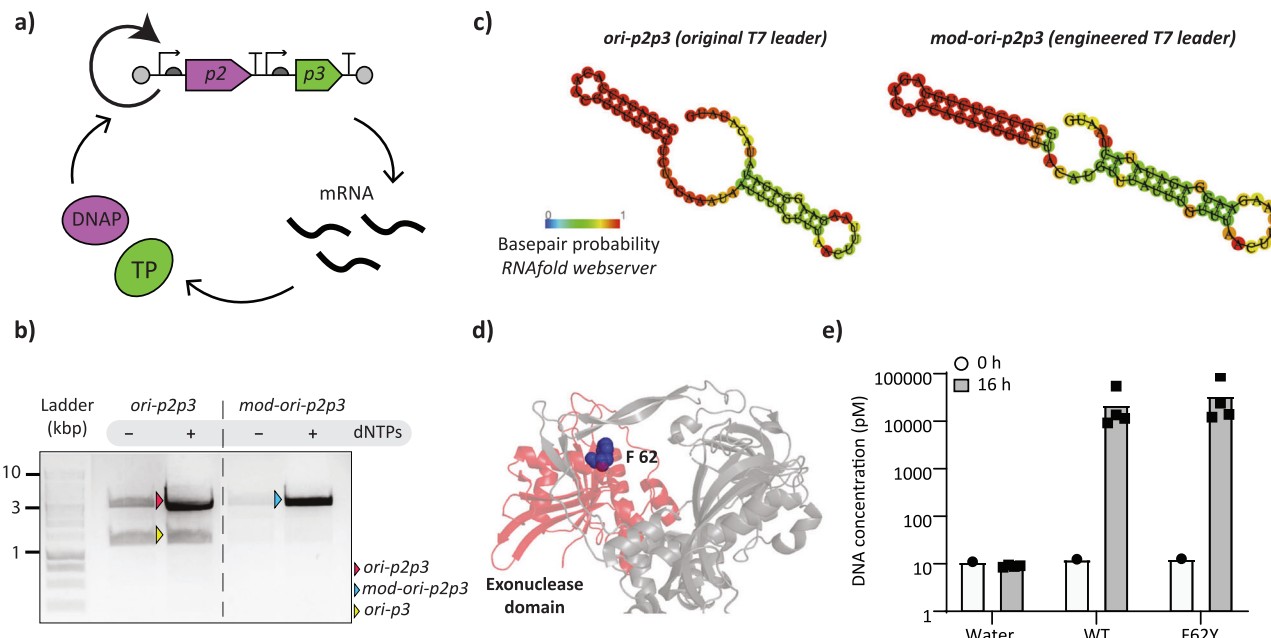

**Fig. 1 | Engineering of a DNA self-replicator. a** Schematic illustration of self-replication by a synthetic DNA replicator encoding DNAP and TP from the Φ29 bacteriophage. Expression in PURE system leads to an autocatalytic network resulting in exponential amplification of the two-gene DNA template. **b** In-liposome IVTTR of original and modified *ori-p2p3* templates assayed by agarose gel electrophoresis of PCR-recovered DNA. **c** Predicted structures of original T7 leader sequences in *ori-p2p3* and an artificial alternative in *mod-ori-p2p3*. RNAfold webserver drawing of minimum free energy plain structure is shown with indicated base pair probabilities. **d** Crystal structure of Φ29 DNAP (PDB 2PY5) with the exonuclease domain highlighted in red and F62 residue in blue. The adapted protein structure was generated with PyMOL Molecular Graphics System. **e** Assay of *ori-pssA* amplification in bulk IVTTR reaction by DNAP (WT or F62Y) and TP expressed from non-replicating circular template. The concentration of *ori-pssA* DNA was quantified by qPCR before and after (16 h) IVTTR, showing similar amplification levels for both polymerases. Individual symbols correspond to data from biological replicates (independent experiments). Source data are provided as a Source Data file.

compartmentalised inside giant unilamellar vesicles, or liposomes. The lipid composition of the vesicles was manufactured to resemble that of the *E. coli* inner membrane[7]. IVTT-supported self-replication was tested in two conditions: "bulk IVTTR", wherein the reaction is conducted in the absence of compartmentalization, and "in-liposome IVTTR", wherein the reaction components are compartmentalized in liposomes, and self-amplification outside of liposomes is prevented by the addition of DNase I in the external solution.

First, we aimed to optimize the sequence of our original template *ori-p2p3* for long-term evolutionary experiments. During bulk (Supplementary Fig. 1b) and in liposome (Fig. 1b) IVTTR, a main self-replication product of size 3.2 kb was generated, as well as an unexpected additional band of size 1.4 kb. With the concern that this fragment could be a self-produced molecular parasite[39] that could significantly hinder an evolutionary experiment, we decided to explore the nature of this shorter fragment and find possible ways to prevent its re-appearance (Supplementary Note 1). Sanger sequencing of the shorter fragment revealed that it formed via recombination at the repeated leader sequence upstream of each gene. We thus engineered a modified *ori-p2p3* template, called *mod-ori-p2p3*, with an artificial T7 leader sequence upstream of the *p3* gene. The leader was designed to form a hairpin RNA structure (Fig. 1c), which was found to be important for gene expression in this system (Supplementary Note 1). Expression in PURE system resulted in a similar yield of TP (Supplementary Fig. 1a) and self-replication ability as the original *ori-p2p3* both in bulk (Supplementary Fig. 1b) and in liposomes (Fig. 1b). Notably, *mod-ori-p2p3* produces less of the smaller 1.4 kb-product (Fig. 1b), making it a better template for our in vitro evolutionary experiments.

Next, we reduced the set of proteins for self-replication of *mod-ori-p2p3* in liposomes by omitting DSB without compromising IVTTR efficiency (Supplementary Note 2, Supplementary Fig. 2). We also wondered if it was possible to modulate the rate of evolution by using a nuclease-deficient variant of Φ29 DNAP (Supplementary Note 3). We thus explored the F62Y mutation (Fig. 1d) that was reported to considerably reduce the exonucleolytic activity of DNAP and increase the frequency of nucleotide misincorporation[61]. After validating TP-primed DNA amplification activity by this DNAP variant in a bulk IVTTR reaction using a heterologous DNA (Φ29 origin-flanked unrelated gene *pssA*) (Fig. 1e), we showed that self-replication activity of *mod-ori-p2(F62Y)p3* in liposomes is similar to that using *mod-ori-p2p3* as the template (Supplementary Fig. 1c).

We then demonstrated that our in-liposome IVTTR method is viable for in vitro self-selection of DNA replicators by performing a mock selection experiment (Supplementary Note 4, Supplementary Fig. 3). The selection principle of active (or more active) self-replicators would be based on their ability to clonally and differentially amplify within individual liposomes (Supplementary Fig. 3a), thus outcompeting less active variants based on Darwinian principles[46,59,62]. Briefly, Φ29-origin-flanked unrelated gene *plsB* was spiked with 2% molar ratio of *mod-ori-p2p3*. After a single round of in-liposome IVTTR, the fraction of *mod-ori-p2p3* in the mixture increased 10-fold (Supplementary Fig. 3b), suggesting that this system can support in vitro Darwinian evolution of a DNA-encoded replicating system.

## Evolution of self-replicators over multiple rounds of intermittent evolution

To develop a Darwinian synthetic protocell, we first sought to establish a standard protocol for in-liposome evolution of DNA replicators. In our initial experiment, the DNA template is encapsulated in liposomes, incubated overnight at 30 °C to complete the compartmentalized IVTTR reactions, after which the self-amplified DNA is extracted from the liposomes, amplified by PCR, and re-encapsulated in fresh liposomes with PURE and DNA replication additives for the next round of evolution. We dubbed this scheme "Intermittent" evolution, since DNA

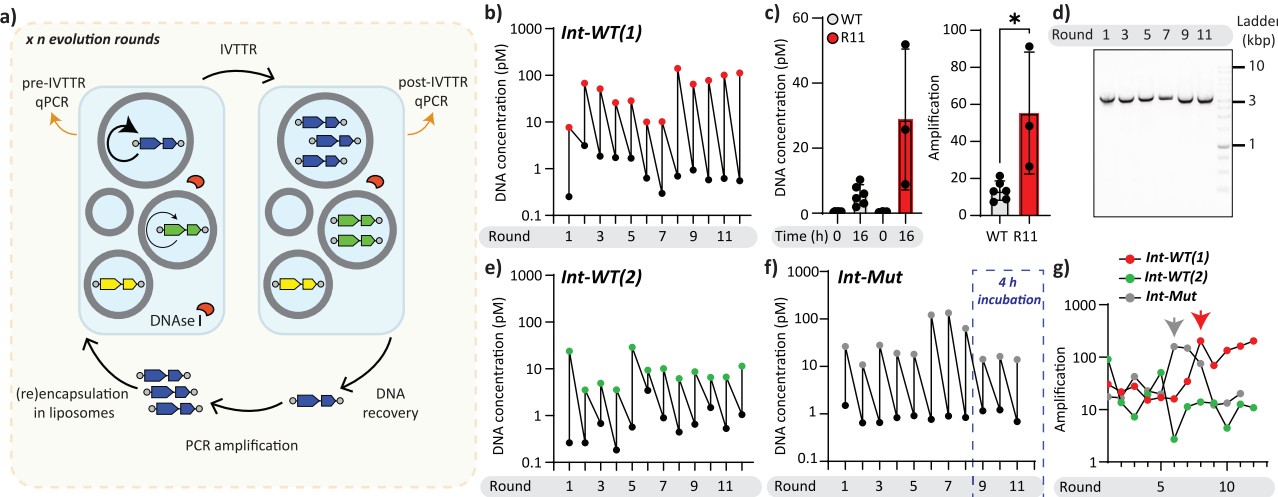

**Fig. 2 | Intermittent evolution of *mod-ori-p2p3* and *mod-ori-p2(F62Y)p3*.**
**a** Schematic illustration of the experimental set-up for the intermittent evolution campaign of DNA self-replicators. **b** Trajectories of *mod-ori-p2p3* concentrations in liposomes in the intermittent evolutionary campaign, Int-WT(1). DNA was quantified by qPCR using primers that target a region in the *p2* gene. **c** Comparison of in-liposome IVTTR in WT and pooled DNA population from Int-WT(1) at evolution round 11 (R11). Data are presented as mean (bar height) ± standard deviation values from three to six biological replicates. *$P < 0.05$. **d** Size analysis of IVTTR and PCR-amplified DNA during Int-WT(1) by agarose gel electrophoresis. **e** Trajectories of *mod-ori-p2p3* concentrations in liposomes in the intermittent evolutionary campaign, Int-WT(2). **f** Trajectories of *mod-ori-p2(F62Y)p3* concentrations in liposomes in the intermittent evolutionary campaign, Int-Mut. The last three evolution rounds, where IVTTR incubation time was reduced from 16 to 4 h, are highlighted with a blue dashed line. **g** DNA amplification trajectories of all three evolutionary campaigns. The arrows indicate the evolution round at which the replication ability improved. Source data are provided as a Source Data file.

is amplified by PCR between rounds of evolution (Fig. 2a). The upside of this scheme is that the recovered DNA can be controlled for the desired size (by agarose gel electrophoresis and gel extraction) and concentration ($A_{260}$-based quantitation before re-encapsulation) at each round of evolution. Moreover, as long as PCR amplification is successful, the possibility of self-replicators gradually going extinct is minimized. The downside of this scheme is the reliance on PCR between rounds of evolution and the possibility of PCR preferentially amplifying shorter variants with no self-replication activity.

We asked if repeated cycles of intermittent experimental evolution would result in the DNA (i) gradually losing its ability to self-amplify due to the accumulation of non-replicating variants (extinction of viable replicators) (ii) retaining its initial self-replication activity at the same level (neutral drift), or (iii) gradually improving its self-replication activity (adaptive evolution). We started our evolutionary campaign with *mod-ori-p2p3* linear PCR fragment. At each round of in vitro evolution, we encapsulated in liposomes the IVTTR reaction mix (no DSB added) along with 10 pM DNA (expected number of molecules per liposome, or $\lambda = 0.2$, Supplementary Note 4). Amplification reaction outside of liposomes was prohibited by adding DNase I after vesicle formation. Clonal amplification of self-replicators was performed at 30 °C for 16 h, after which the DNase I was thermally inactivated. The DNA was then released from the vesicles by an osmotic shock (dilution in water) and the pooled replicator population was further amplified using conventional PCR. The expected *mod-ori-p2p3* size (~ 3.2 kb) was verified by agarose gel electrophoresis, and the full-length *mod-ori-p2p3* DNA band was gel-purified to limit the possible propagation of molecular parasites to the next IVTTR round. The resulting DNA was carried on to the next round of evolution and encapsulated again at $\lambda = 0.2$. The sequence diversity was allowed to accumulate passively during IVTTR (expected $10^{-5}$ to $10^{-6}$ substitutions/base/doubling[63–65] for Φ29 DNA polymerase) and PCR amplification between rounds ($0.7–1.2 \times 10^{-5}$ substitutions/base/doubling for KOD DNA polymerase[66,67]).

We performed 12 rounds of intermittent in-liposome evolution, and called this evolutionary campaign Int-WT(1). We quantified the initial and final amounts of DNA at each round and discovered that

within 12 rounds of in vitro evolution, the amplification of self-replicating DNA improved at least 5-fold (Fig. 2b,c,g). The length of the amplified DNA did not change over the course of evolution, and no additional DNA products were observed (Fig. 2d). We repeated Int-WT(1) in a separate evolutionary experiment (Int-WT2), this time without gel purification since inspection of PCR product size by gel electrophoresis did not reveal additional bands other than full-length *mod-ori-p2p3* (Supplementary Fig. 4). The DNA amplification profile of this experiment once again confirmed persistent self-amplification of the DNA replicator throughout the evolution campaign (Fig. 2e,g). This time, however, we observed no improvement (nor deterioration) of DNA replication efficiency over evolution rounds.

To investigate the impact of decreased proofreading in F62Y variant of Φ29 DNAP[61] on the evolutionary dynamics, we applied the same protocol for in vitro evolution of *mod-ori-p2(F62Y)p3* variant, and called this campaign Int-Mut. In this experiment, the amplification of self-replicating DNA improved within 10 rounds of evolution (Fig. 2f,g). By the 9th round of evolution, we increased the selection stringency of Int-Mut evolution by reducing the IVTTR incubation time from 16 hours to 4 h at 30 °C. As a result, the replication yield dropped in the last 3 rounds of evolution, after which we stopped the evolution campaign. Subsequent NGS analysis of Int-Mut suggested that a contamination event from Int-WT(1) evolution affected the course of evolution of Int-Mut (Section Emergence of DNA variants and fixation dynamics). Overall, we conclude that compartmentalised, transcription-translation-coupled self-replication of DNA using an intermittent evolution protocol is compatible with the survival of functional DNA replicators. In two instances over three independent evolution campaigns, the DNA replicator self-amplification ability improved within only 10 rounds of evolution (Fig. 2g).

## Semi-continuous evolution of *mod-ori-p2p3*
Next, we investigated whether it is possible to minimize the researcher intervention in the in vitro evolution process via a more streamlined evolution protocol. In such a system, amplified DNA from a fraction of liposomes would be recursively passed on to an excess of fresh liposomes via vesicle fusion and fission. This scheme would obviate out-of-

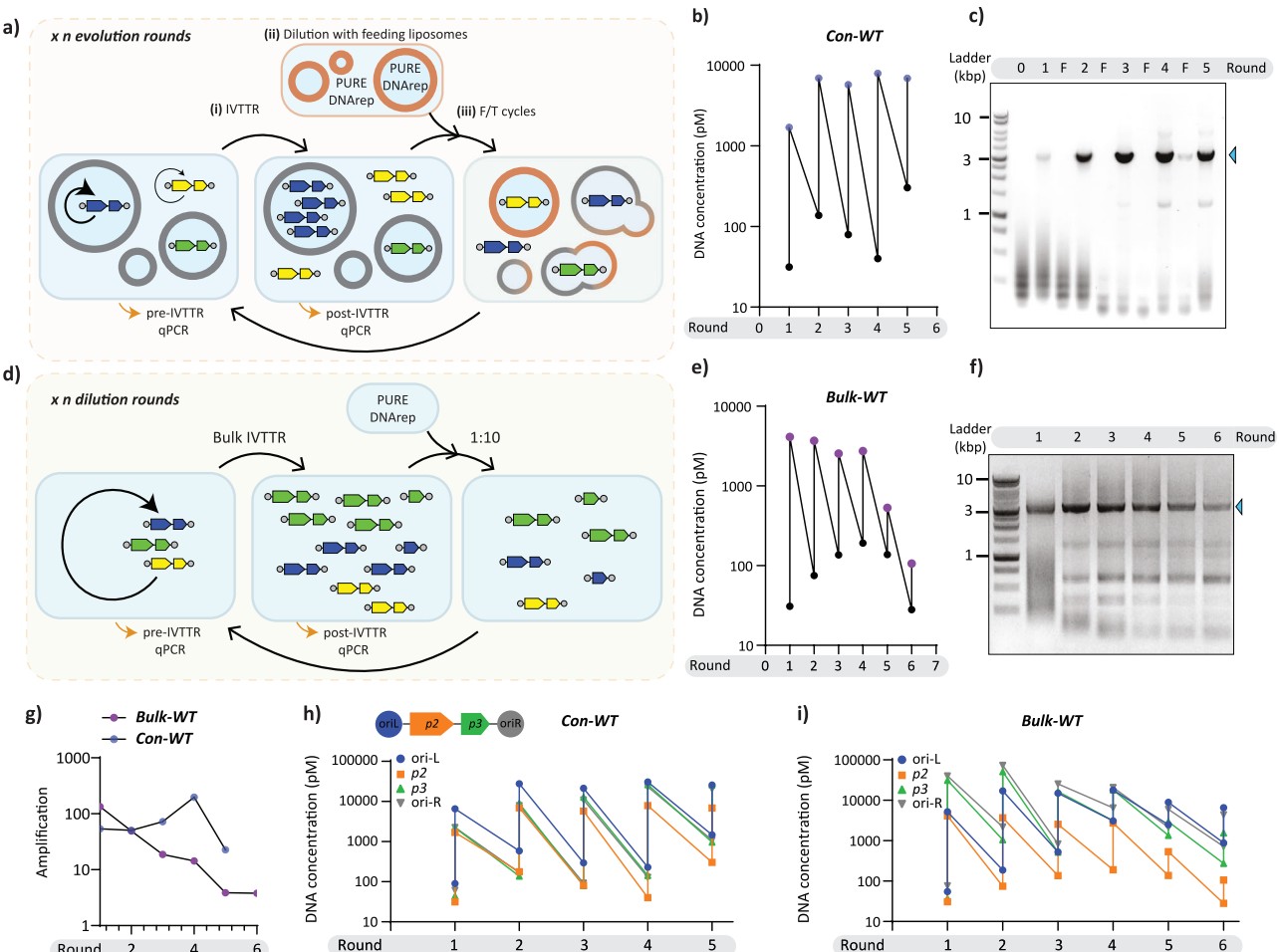

**Fig. 3 | Semi-continuous evolution of *mod-ori-p2p3*. a** Schematic illustration of the experimental set-up for a semi-continuous evolution approach in liposomes. Step (i) in-liposome IVTTR (ii) 100-fold dilution of old vesicle suspension in a suspension of fresh vesicles, (iii) liposome fusion-fission promoted by cycles of F/T. **b** Trajectories of *mod-ori-p2p3* concentrations in liposomes in the semi-continuous evolutionary campaign, Cont-WT, as measured by qPCR. Each IVTTR was incubated for 16 h and liposomes were diluted 100 times with feeding vesicles. The target region for qPCR quantification ( ~ 200 bp) belongs to the *p2* gene. **c** Size analysis of PCR-amplified DNA during Con-WT by agarose gel electrophoresis. The arrowhead indicates the full-length replicator. **d** Schematic illustration of the experimental set-up for the bulk serial transfer campaign (Bulk-WT). Bulk IVTTR reactions were incubated for 16 h. The next round of IVTTR was started after 10-fold dilution of the

pre-ran IVTTR reaction in a fresh PURE system complemented with DNA replication factors. **e** Trajectories of *mod-ori-p2p3* concentrations in a bulk reaction, Bulk-WT, as measured by qPCR. **f** Size analysis of PCR-amplified DNA during Bulk-WT by agarose gel electrophoresis. The arrowhead indicates the full-length replicator. **g** Amplification of DNA in the two evolution campaigns. **h,i** Quantitative comparison of the abundance of different DNA regions throughout the evolutionary rounds in Con-WT (**h**) and Bulk-WT (**i**). DNA presence was assayed by quantitative PCR using standard curves for each primer pair. The inset cartoon is a schematic illustration of *mod-ori-p2p3* self-replicator regions that were targeted by qPCR. Full-length replicator persists in liposome but is outcompeted by DNA that does not contain *p2* gene in bulk IVTTR. Source data are provided as a Source Data file.

liposome PCR amplification and controlled re-encapsulation of DNA (Fig. 3a). Fresh feeding vesicles supply additional lipids, PURE components, and additives for DNA replication. Both fusion and fission events are promoted by freeze-thaw (F/T) cycles, during which pooling and stochastic redistribution of the DNA content are expected. A similar protocol was used by Tsuji et al. for the replication of RNA over multiple rounds of liposome cultivation[60]. We called this evolution scheme "semi-continuous", as it is a step towards continuous in vitro evolution, where a synthetic cell can pass on its genetic information to the next generation without researcher intervention[5]. Noteworthily, the fact that the output of the selection process is exactly the same molecule as the input is a critical feature to enable continuous evolution.

Figure 3a summarizes the main steps of the semi-continuous evolution cycle: (i) in-liposome IVTTR, (ii) dilution with a solution containing feeding vesicles (same composition as the 'self-replicator vesicles' except that DNA was omitted), (iii) application of a F/T cycle

to promote liposome fusion-fission, thereby releasing and stochastically re-entrapping the DNA content for the next round of evolution. We reasoned that the transfer of genetic information coupled to a new round of IVTTR would enable propagation of the self-replicator if DNA amplification overcompensates for the dilution effect caused by the addition of the feeding vesicles[38,41].

First, we confirmed that F/T cycles led to membrane and liposome content mixing (Supplementary Fig. 5). DNA leakage into the outer solution during F/T was estimated to be 50% (Supplementary Fig. 6a,b), so around half of amplified DNA would still remain inside liposomes (old and fresh). Semi-continuous evolution was realized by incubating the IVTTR reactions for 16 h at 30 °C and diluting the liposomes 100-fold between rounds. This experiment, which we dubbed Con-WT, was conducted in the presence of DSB to maintain the complete pool of replicating DNA variants by enabling outside-of-liposome IVTTR. DNA that replicated in the interior of liposomes but leaked out during F/T (about 50%, see Fig. S6a,b) can potentially be

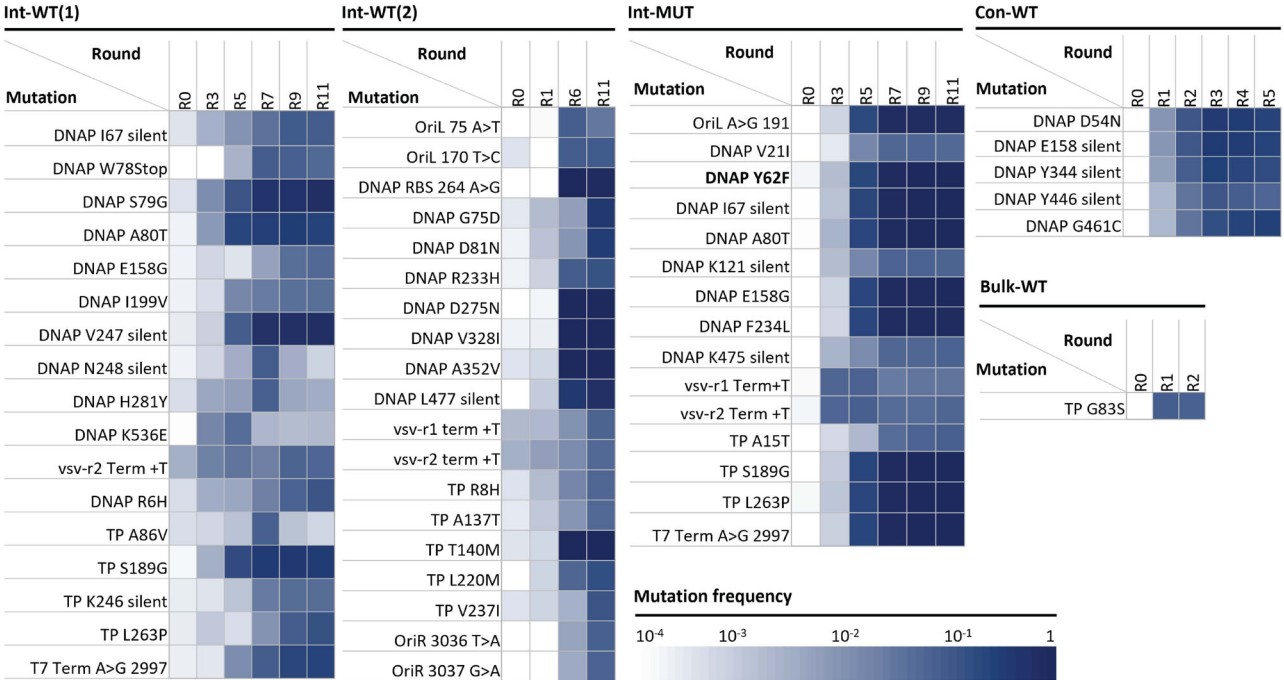

**Fig. 4 | Analysis of evolutionary patterns.** Heatmap of mutation frequencies enriched to at least 5% at any round of in vitro evolution. Int-WT(1), trial 1 of intermittent in-vesiculo evolution on a starting WT (codon-optimized) sequence of *mod-ori-p2p3*. Int-WT(2), trial 2 of intermittent in-vesiculo evolution on a starting WT (codon-optimized) sequence of *mod-ori-p2p3*. Int-Mut, intermittent in-vesiculo evolution on a starting F62Y variant of codon-optimized sequence *mod-ori-p2p3*. Con-WT, semi-continuous in-vesiculo evolution on a starting WT (codon-optimized) sequence *mod-ori-p2p3*. And, Bulk-WT, serial transfer of bulk IVTTR reaction starting with WT (codon-optimized) sequence *mod-ori-p2p3*.

amplified and re-encapsulated in a subsequent round, hence propagate, maintaining the pool of variants that would otherwise be washed out by dilution if DNA replication outside liposomes was disabled (no DSB). Moreover, the addition of external DNase I, used to prohibit external DNA amplification in the intermittent evolution experiment, was avoided here due to the risk of its entrapment in liposomes during F/T between rounds of evolution. Of note, external DNA amplification may be less effective than that of internal due to the beneficial effects of molecular crowding and confinement on gene expression inside of liposomes (Supplementary Fig. 6c–g). Further, transient compartmentalization may be sufficient to sustain replication and adaptive evolution[68,69].

We found that DNA replicators persisted over at least five cycles (Fig. 3b). Accumulation of the full-length replicator in the course of evolution was verified by running PCR-amplified DNA (using ori-binding primers) from each round on an agarose gel (Fig. 3c), suggesting that continuous DNA evolution is possible in the presence of liposomes. In contrary, serial dilution of *mod-orip2p3* with fresh PURE system and DNA replication components in the absence of liposomes (Fig. 3d) resulted in a gradual decrease of DNA concentration and self-replication was totally suppressed at round 6 (Fig. 3e,g). In this evolutionary experiment, called Bulk-WT, the dilution factor was set to 10-fold to maintain a sufficient amount of DNA for the next round. Agarose gel analysis of DNA samples after full-length recovery PCR revealed the presence of short replication products already at round 2, while the amount of the full-length replicator gradually diminished (Fig. 3f).

Quantitative PCR targeting multiple regions distributed over the entire length of *mod-ori-p2p3* was performed directly from diluted samples before and after each round of the in-liposome and bulk evolution experiments (Supplementary Fig. 7, Fig. 3h,i). In the in-liposome Con-WT campaign, the *p2* gene driving replication follows the dynamic pattern of the other targeted regions (Fig. 3h), showing persistent survival (i.e., the self-replicator has not been washed away or outcompeted

by short replicons) of the full-length DNA self-replicator in the presence of liposomes. In contrast, the abundance of *p2* in the bulk evolution experiment decreased faster than the other regions (Fig. 3i) suggesting that shorter parasites outcompeted the self-replicator. The takeover of molecular parasites and extinction of the full-length replicator in Bulk-WT suggests that liposomes enable sustained propagation through Darwinian selection that arises due to the genotype-phenotype link provided by (transient) compartmentalisation.

To ascertain that the different persistency of Con-WT and Bulk-WT did not result from different dilution factors (100- and 10-fold dilutions for Con-WT and Bulk-WT, respectively), we repeated both experiments this time by changing the dilution factor (Supplementary Note 5). The results corroborate our finding that the DNA self-replicator persists longer in the presence of liposomes than in bulk reactions (Supplementary Figs. 8,9).

Overall, persistence of a DNA self-replicator is experimentally demonstrated in both intermittent and semi-continuous evolutionary settings. We next sought to determine which genetic variations were acquired that may have conferred a selection advantage.

## Emergence of DNA variants and fixation dynamics

To investigate the evolutionary processes that took place during the campaigns Int-WT(1), Int-WT(2), Int-Mut, and Con-WT, we deep-sequenced the PCR-amplified products of IVTTR at different evolution rounds using Illumina next-generation sequencing (NGS) technology after random fragmentation. We mapped and extracted the frequency of occurrence of all the point mutations that were detected at a frequency of at least 1% (Supplementary Data 1) or 5% (Fig. 4, Supplementary Fig. 10) in at least one of the evolutionary rounds. We found that in the evolving Int-WT(1) population, some of the mutations increased in frequency earlier in the rounds and decreased in the later rounds, while some of the others increased in frequency and became dominant in the later rounds (Fig. 4). In particular, the nonsynonymous mutations S79G and A80T in the *p2* (DNAP) gene reached 67% and 26%

frequencies by round 11 in the evolutionary campaign Int-WT(1). These two mutations seem to have appeared and been selected independently of each other, as they are never jointly observed on sequencing reads. Sanger sequencing results of twelve single clones isolated from round 11 of Int-WT(1) confirmed that S79G (found in ten clones) and A80T (found in one clone) mutations were present independently (Supplementary Table 1). Additionally, eight of the clones harbouring the S79G mutation also carried a V247V silent mutation (Supplementary Table 1), which was found at a frequency of at least 5% by NGS analysis (Fig. 4). In Int-WT(2), NGS data show an accumulation of a different set of nonsynonymous mutations (Fig. 4). We also observed instances of evolutionary convergence, as S79G and A80T were also detected in Int-WT(2), although at a lower frequency (hence not shown in Fig. 4).

In the Int-Mut evolution experiment, a similar pattern of genetic diversification was observed throughout the evolution. However, we also observed a rapid and simultaneous takeover of a set of nine mutations, which became dominant by round 7 (87–90%). These included mutations in the *p2* gene Y62F (double mutation and reversal to WT from F62Y), A80T, I67 silent, E158G, F234L, in the *p3* gene S189G and L263P, and in the intergenic region 2997 A > G (T7 terminus) (Fig. 4, Supplementary Fig. 11). This pattern suggests a contamination event from one of the later rounds of the Int-WT(1) evolution into the Int-Mut evolution, since all of these mutations can be found in the Int-WT(1) dataset, although they never reached allele frequencies this high in Int-WT(1). Unfortunately, the early-in-the-evolution replacement of the F62Y mutation by the WT contaminant meant that we could not reliably analyse its contribution to the evolution's course. Remarkably though, in Int-Mut evolution lineage, the mutation S79G also appears and gets selected among variants still containing the F62Y (although those are gradually being outcompeted, Supplementary Fig. 11). This strongly supports the positive contribution of this substitution.

Two silent mutations in the *p2* gene K121 and K475, both found only in Int-Mut at max frequency of 6.8% at R9 and R11 in both cases, are worthy of note (Fig. 4). In both situations, we have two consecutive lysine residues, which together are encoded by 6 A's in a row. Both of these silent mutations are AAA/AAG mutations in the first of the two lysine codons that disrupted the homopolymer runs of 6 A's. Repeated nucleotides are known to be a source of DNA polymerase-mediated frame-shift mutations in coding sequences[70], thus making them potential hubs for deleterious mutational hotspots. We hypothesize that these homopolymeric runs could act as local sources of genetic instability that would result in out-competition by a more stable, although synonymous replicator. However, as it was also previously reported that in consecutive lysine sequences, homopolymeric A stretches can result in ribosome sliding and poorer translation[71], we cannot completely exclude protein expression level effect[71]. Interestingly, the AAG codon at positions 475 and 121 are both AAG in the original Φ29 genome and were changed to the more frequent lysine codon AAA by the codon-optimization algorithm during generation of the parental *ori-p2p3* construct (Supplementary Table 2), but in vitro evolution changed them back to their native sequence.

Other examples of such a codon reversal include I67 synonymous mutation, found in both Int-WT(1) and Int-Mut at maximum frequencies of 9.6% and 9.7%, respectively. Here, the ATT codon was reverted to ATC, which is originally present in the Φ29 genome and has a lower codon frequency (Supplementary Table 2). Mutations K475K, K121K, and I67I, are examples indicating that the genetic diversity in Int-Mut accumulated partly due to contaminating DNA from Int-WT(1), and partly independently of Int-WT(1), generating a unique evolutionary path. In Con-WT, a synonymous mutation in *p2* gene restored the original Φ29 codon sequence (Y344Y with TAC→TAT) (Supplementary Table 2). However, in this case, TAT is a more frequent codon than TAC in *E. coli*. Synonymous mutations in the coding sequences may regulate protein expression profiles or even protein folding by

controlling local translation rate[72]. However, we cannot exclude the hypothesis that these synonymous mutations can be examples of passenger mutations, enriching only because they are associated with another beneficial mutation.

Some other enriched mutations in both Int-WT(1), Int-WT(2), and Int-Mut included additional T's in the polyT stretches of vsv-r1, vsv-r2, and T7 terminators. This could be another example of a mutational hotspot due to DNA polymerase slipping[70], which may have been enriched due to improved transcription termination[73].

Larger rearrangement of the replicator can also be observed in the two intermittent evolution campaigns. In both cases, a modified right origin of replication, adopting the extreme 24 bases from the left origin, is dominant in sequences of the latter rounds (Supplementary Figs. 12–14). Although this swapping may be promoted by some level of homology between the two extremities and the change correspond to the PCR primer length, the mechanism by which it gets selected is unclear. In addition, in the Int-WT case, a large deletion of around 200 bases, preserving the ORF but removing most of the right origin can be spotted from round 5 and become dominant in later rounds (Supplementary Figs. 13,14).

To estimate the accumulation of genetic diversity throughout evolution, we integrated all the frequencies of mutations that were found above a certain threshold to approximate an average number of mutations found per single DNA molecule in the population. The frequency threshold was set to 0.1% considering the median Phred score of the sequencing run (35, corresponding to an estimated 0.03% error rate). We found that the average number of mutations per DNA molecule gradually increased from 0 before the start of evolution, to 4–8 by round 11 of Int-WT(1) and Int-WT(2), Int-Mut (Supplementary Fig. 15a). These results suggest that the mutational load in our experiment was enough to generate sufficient genetic diversity for selection to take place. We also estimated the total number of positions mutated in the entire DNA population, and observed that the number increased from 0 to over 600 by R11 of Int-WT(1) and Int-WT(2), and over 800 of Int-Mut (Supplementary Fig. 15b).

Notably, the mutation accumulation rate was much lower for the in vesiculo Con-WT experiment, plateauing at around 1 mutation/molecule after round 3, whereas the intermittent evolution method kept accumulating mutations throughout 10 rounds of evolution (Supplementary Fig. 15a). Total mutated positions per evolution round was also much lower for the continuous evolution method (Supplementary Fig. 15b). These data suggest that PCR recovery used between rounds of mutation in Int-WT produced a significant fraction of the genetic diversity. Therefore, using a mutator DNAP variant might be more beneficial for faster accumulation of mutations in semi-continuous evolution. Another possibility is that the phenotype-genotype link in semi-continuous scheme is weaker, slowing down the fixation of improved variants[51].

## Characterization of enriched variants

We next characterized some of the most enriched missense mutations in Int-WT(1) and Int-Mut. To assess whether mutations S79G and A80T in the *p2* gene were sufficient to improve DNA amplification, we created single and double mutants starting from the parental template and subjected them to in-liposome and bulk IVTTR. In-liposome IVTTR of the single-mutant constructs led to a significant ($P < 0.05$) improvement in the final DNA yield compared to the parental template (*mod-ori-p2p3*), while the enhancement of the combined mutations (S79G-A80T) was only moderate (Fig. 5a,b). Since similar amplification increase was observed in R11 of Int-WT(1) compared to the parental sequence (Fig. 2c), we conclude that S79G or A80T mutations accounted for the majority of the self-amplification improvement observed in the population. Moreover, in-liposome IVTTR kinetics demonstrated a significant ($P < 0.05$) difference in self-replication rate for S79G variant DNA when compared to the parental template (Fig. 5c).

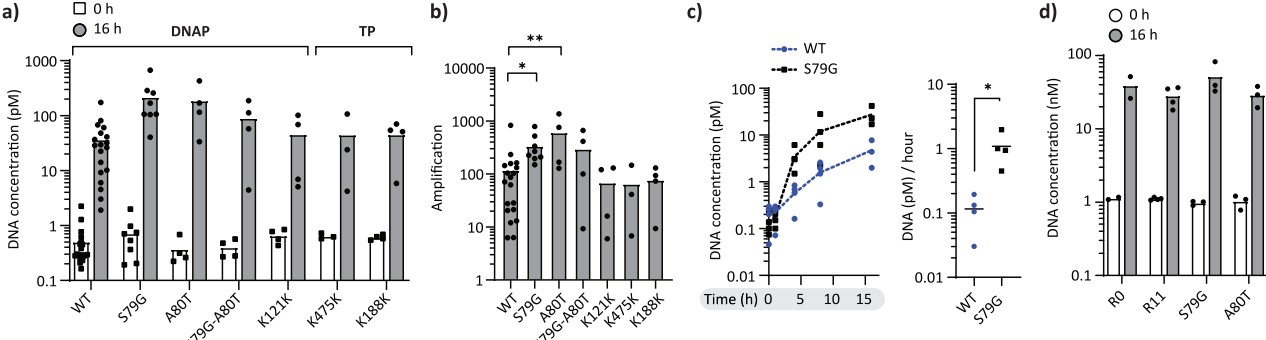

**Fig. 5 | Reverse engineering and characterization of fixed end-point mutations.** **a** Assessment of reverse-engineered self-replicator variants' replication activity in in-liposome IVTTR by qPCR. qPCR amplicon is amplified from a *p2* gene region of *mod-ori-p2p3*. **b** Self-amplification displayed as ratio of DNA concentration at 16 h to initial template concentration at 0 h in panel a. *$P < 0.05$, **$P < 0.01$. **c** Left: Comparison of in-liposome IVTTR kinetics of the parental *mod-ori-p2p3* DNA template and its variant with the S79G mutation in the *p2* gene. Absolute quantification of DNA was performed by qPCR ($n = 3$ biological replicates). Dashed lines connect the mean values of replicates. Right: Apparent maximum DNA replication rates defined as the highest slopes (between 1 and 4-h time points) in the kinetic curves. *$P < 0.05$. **d** Comparison of parental *mod-orip2p3* (R0), recovered DNA from Int-WT(1) round 11, and reverse-engineered self-replicator variants' replication activity in bulk IVTTR by qPCR. Data points are from three to 19 biological replicates, except for panel d, where condition R0 was repeated twice. Source data are provided as a Source Data file.

Under bulk IVTTR conditions starting with higher DNA concentrations (1–2 nM), no differences in DNA replication were observed between the parental DNA (round 0 PCR of *mod-ori-p2p3*), the reverse-engineered *mod-ori-p2p3* template harbouring the S79G or A80T mutations, and PCR-recovered DNA from round 11 on Int-WT(1) (Fig. 5d). This finding may be a result of a mutational fitness advantage in response to compartmentalisation in liposomes, where macromolecular crowding, confinement, or membrane effects could play a role. These results support the idea that our self-replicating system underwent evolutionary adaptation to the specific in-liposome IVTTR condition.

Further characterization of single variants did not reveal the exact mechanism by which this improvement is achieved. End-point in vitro protein expression assays with Green-Lys in vitro translation labelling system and liquid chromatography-mass spectrometry showed no improvement in the amounts of synthesized proteins (Supplementary Figs. 16 and 17). To disentangle protein property from DNA template effects, we characterized the purified S79G, A80T, and the double mutant DNAP variants. Although all three protein variants were indeed active polymerases, none of them exhibited a higher ability than the WT to replicate DNA in bulk IVTT reactions, both in protein-primed and DNA-primed settings (Supplementary Note 6, Supplementary Figs. 18 and 19). Hence, it is unlikely that the amino acid residue substitution in the translated DNAP protein improves its replication activity. In any case, the opposite effects of these mutations in different conditions suggest that the evolution experiments resulted in DNA template or protein variants that are more fit only in the specific environment in which they were selected, in agreement with the directed evolution maxim 'you get what you select for'[74].

Reverse engineering of the silent mutations K121 and K475 (DNAP gene), and K188 (TP gene) was also performed. All single mutations led to a similar DNA amplification yield as the parental DNA (Fig. 5a,b), suggesting that any beneficial effects may only become apparent in a richer genetic context, or that they act as hotspot stabilizing mutations without any direct effect on expression or replication.

## Discussion

This work shows that a de novo designed linear DNA self-replicator is capable of undergoing sustainable amplification and adaptability in a synthetic protocellular environment. Our primary goal was to understand broader evolutionary principles and processes that can lead to the emergence of self-replicating, functionally integrated entities, and ultimately a synthetic cell. The discovery that adaptive evolution arose relatively fast (within 10 rounds of evolution) compared to repetitive, concatemeric DNA replication[45], where mutation effects average out due to multiple gene copies per molecule, makes our DNA self-replicating mechanism a good candidate for implementation in an evolving synthetic cell.

It has long been recognized that compartmentalisation is important for the functional selection of self-replicating systems[75]. In particular, spatial organization can prevent the spread of nonfunctional replicons named parasites, resulting in the survival of compartments enriched with self-replicating molecules that would otherwise become extinct as parasites take over. The role of compartmentalisation has been experimentally verified using PCR and RNA replication systems[76]. Moreover, transient compartmentalisation was shown to be sufficient for selecting functional RNA replicators and purging the parasites[38]. Our results suggest that parasite takeover was responsible for the extinction of DNA self-replicators in bulk-WT reactions (Fig. 3e–g,i), demonstrating the importance of compartmentalizing liposomes for sustainable IVTTR.

Our DNA self-replicating scheme enables the emergence and maintenance of genetic diversity in liposome populations such that selection can operate. A number of fixed mutations have been identified. Yet, the exact causes for mutational fitness advantage needs further investigation. Although the fixed mutations S79G and A80T, located in the exonuclease domain of DNAP, appear sufficient to increase the replication ability in liposomes, no improvement of the activity of the purified DNAP variants could be detected. The results indicate that selection for increased self-replication efficiency is specific to the in-liposome IVTTR environment in which the mutations emerged, and that (genetic) effects other than protein property may also contribute to selection. Moreover, the specific factors defining the selection pressure for adaptive evolution remain to be explored. It is clear however that the chimeric nature of the self-replicating system (Φ29 phage replication proteins, T7 bacteriophage RNAP, *E. coli* translation machinery, the synthetic DNA template, artificial liposomes) applies on its own a strong evolutionary pressure. Adjusting the selection conditions (e.g., temperature, lipids, reaction time, PURE composition) to alter replicative fitness can provide insights into the adaptation potential of the DNA replicator.

Considering that the sequence space is strongly reduced compared to living organisms, including nearly minimal bacterial cells[77], it may be easier to understand the first principles of self-replicating systems due to the fewer targets on which positive selection can act. This provides an experimental testbed to evaluate hypotheses on the

fundamental concepts of evolution in living systems and to predict how minimal cells respond to changing situations. Furthermore, our platform can be used to model viral replication of genomic DNA through transcription-translation in bacterial host organisms, as well as the underlying evolutionary mechanisms, with implications in the development of new therapeutic methods. Finally, we envision that integration of more genes in this minimal DNA self-replicator constitutes the next step for co-evolving multiple cellular functions in vitro through Darwinian optimization.

## Methods

### Buffers and solutions

All buffers and solutions were made using Milli-Q grade water with 18.2 MΩ resistivity (Millipore, USA). Chemicals were purchased from Sigma-Aldrich unless otherwise indicated.

### Construction of DNA fragments for IVTTR reactions

Plasmid G340, which contains *mod-ori-p2p3* construct with mutated T7 leader upstream of *p3* gene, was prepared from G95 plasmid (original *ori-p2p3* construct from[7]). The fragment encoding the T7 mutated leader sequence was prepared by primer extension of the overlapping primer pair 1058 and 1060 ChD. The genes *p2* and *p3* were amplified from G95 plasmid using the primer pairs 1049/1056 ChD for *p2*, and 1057/1052 ChD for *p3*. The three fragments were assembled into a KpnI and HindIII-linearized pUC19 vector with Gibson Assembly[78]. Plasmid G371, containing *mod-orip2(F62Y)p3* and encoding for Φ29 DNAP(F62Y), was cloned from G340 plasmid by focused PCR mutagenesis using 948/1132 ChD, and 1131/1137 ChD as primer pairs. The two overlapping DNA fragments were assembled into KpnI/PmeI-linearized G340 plasmid using the Gibson Assembly method[78]. Reverse engineered plasmids containing point mutations enriched over the evolutionary campaigns (G559 for DNAP(S79G), G570 for DNAP(A80T), G569 for DNAP(S79G&A80T), G560 for DNAP(K121K), G561 for DNAP(K475K), and G562 for TP(K188K)), were constructed by mutagenesis PCR utilizing G340 as a DNA template. After PCR, the reactions were treated with Dpn1 for digesting the parental G340 DNA template. The primer pairs used for each DNAP and TP mutagenesis PCR can be found in Supplementary Table 3. All the plasmids were cloned by heat-shock transformation of *E. coli* Top10 strain, and plasmids were extracted from individual cultures outgrown in ampicillin containing LB using Promega PURE yield Plasmid Miniprep kit. Individual clones were screened and confirmed by Sanger sequencing.

To prepare linear *mod-ori-p2p3* DNA fragments for IVTTR experiments, a PCR was performed with phosphorylated primers 491 and 492 ChD. Reactions were set up in 100 μL volume, 500 nM each primer, 200 μM dNTP, ~10 pg/mL DNA template, and 2 units of Phusion High-Fidelity DNA Polymerase (NEB) in HF Phusion buffer. Thermal cycling was performed as follows: 98 °C 30 s initial denaturation, 20 cycles of (98 °C for 5 s, 72 °C for 3 min.), and final extension at 72 °C for 5 min. Extra care was taken to not over-amplify the DNA by too many thermal cycles, as it was found to adversely affect the quality of purified DNA. The amplified PCR fragments were purified using Qiagen QIAquick PCR purification buffers and Qiagen RNeasy MinElute Cleanup columns using the manufacturer's guidelines for QIAquick PCR purification, except for longer pre-elution column drying step (4 min. at 10,000 g with open columns), and elution with 14 μL MilliQ water in the final step. The purified DNA was quantified by Nanodrop 2000c spectrophotometer (Isogen Life Science) and further analysed for size and purity by agarose gel electrophoresis.

### Bulk IVTTR

Bulk replication reactions were set up in PURE*frex* 2.0 (GeneFrontier). A 20-μL reaction consisted of 10 μL solution I, 1 μL solution II, 2 μL solution III, 20 mM ammonium sulphate, 300 μM dNTPs, 375 μg/mL purified Φ29 SSB protein, 105 μg/mL purified Φ29 DSB protein, and 0.6

units/μL of SUPERase·In RNase inhibitor (Themo Fisher), and template DNA at the indicated amount. Reactions were incubated in a nuclease-free PCR tube (VWR) in a Thermal Cycler (C1000 Touch, Biorad) at a default temperature of 30 °C. Incubation time was indicated when appropriate, variating from 4 to 16 h. To analyse the reactions by gel electrophoresis, 10 μL reaction was treated with 0.2 mg/mL RNase A (Promega), 0.25 units RNase One (Promega) at 30 °C for 1–2 hours, followed by 1 mg/mL Proteinase K (Thermo Scientific) at 37 °C for 1–2 hours, and column-purified using the QIAquick PCR purification buffers (Qiagen) and RNeasy MinElute Cleanup columns (Qiagen) using the manufacturer's guidelines for QIAquick PCR purification, except for longer pre-elution column drying step (4 min at 10,000 g with open columns), and elution with 14 μL MilliQ water in the final step. A fraction (generally 6 μL) of the eluate was mixed with an equal volume of 6x purple gel loading dye (NEB) and loaded in 1% agarose gel with ethidium bromide, following which DNA was separated using an electrophoresis system (Bio-Rad). The BenchTop 1-kb DNA Ladder (Promega) was used to estimate the size of DNA.

### Lipid-coated bead preparation

The procedure was adapted from[7] with minor modifications. To prepare lipid-coated beads, a lipid mixture consisting of DOPC (50.8 mol %), DOPE (35.6 mol%), DOPG (11.5 mol%), cardiolipin (2.1 mol%), DSPE-PEG(2000)-biotin (1 mass%) and DHPE-TexasRed (0.5 mass%) for a total mass of 2 mg and 25.4 μmol of rhamnose (Sigma–Aldrich) dissolved in methanol was assembled in a 5-mL round-bottom glass flask. All lipids were purchased at Avanti Polar Lipids and dissolved in chloroform, except the DHPE-TexasRed membrane dye (Invitrogen). Finally, 600 mg of 212–300-μm glass beads (Sigma-Aldrich) was added to the lipid solution, and the organic solvent was removed by of rotary evaporation at 200 mbar for ~2 h, followed by lipid beads collection, aliquoting, and overnight desiccation in individual 2 mL Eppendorf tubes. The dried lipid-coated beads were stored under argon at −20 °C.

### Intermittent evolution: in-liposome IVTTR

Reactions were set up in PURE*frex* 2.0 (GeneFrontier). A 10-μL reaction consisted of 5 μL solution I, 0.5 μL solution II, 1 μL solution III, 20 mM ammonium sulphate, 300 μM dNTPs, 375 μg/mL purified Φ29 SSB protein, 0.6 units/μL of Superase·In RNase inhibitor (Thermo Fisher), and 10 pM template DNA was prepared in a 1.5 mL Eppendorf tube. To the well-mixed reaction, 5 mg lipid-coated beads, already pre-desiccated for at least 20–30 min before use, were added. The 1.5 mL-Eppendorf tube containing the bead-PURE*frex* mixture was next gently rotated on an automatic tube rotator (VWR) at 4 °C along its axis for 30 min for uniform liposome swelling. The mixtures were then subjected to four freeze/thaw cycles (5 s in liquid nitrogen followed by 10 min on ice). Using a cut pipette tip, 5 mL of bead-free liposome suspension (the beads sediment to the bottom of the tube) was transferred to a PCR tube, where it was mixed with 0.5 units of DNase I (NEB). Reactions were incubated in a nuclease-free PCR tube (VWR) in a Thermal Cycler (C1000 Touch, Biorad) at a default temperature of 30 °C for 20 min (for 0-hour sample), or 4–16 h (whenever indicated), after which the DNase I was heat-inactivated at 75 °C for 15 min.

### Intermittent evolution: DNA recovery

To proceed with another round of in-liposome IVTTR, the liposome suspension was then diluted 100-fold in Milli-Q water. Diluted IVTTR reactions were used as templates for PCR amplification using phosphorylated primers 491 and 492 ChD. For this, reactions were set up in 100 μL volume, 300 nM each primer, 400 μM dNTPs, 10 μL diluted liposome suspension, and 2 units of KOD Xtreme Hotstart DNA polymerase in Xtreme buffer. Thermal cycling was performed as follows: 2 min at 94 °C for polymerase activation, and 25–30 cycles of (98 °C for 10 s, 65 °C for 20 s, 68 °C for 1.5 min). Extra care was taken to not over-amplify the DNA by too many cycles, as it was found to negatively

affect DNA recovery during the next round of evolution. The amplified PCR fragments were size-separated on a 0.7–1% agarose gel containing SYBR Safe by gel electrophoresis. Whenever having additional bands on the gel electrophoresis (even if slight) that did not correspond to the expected *mod-ori-p2p3* band size (~ 3.2 kb), the DNA band with the expected size was excised and purified using QIAquick Gel Extraction Kit buffers (Qiagen) and RNeasy MinElute Cleanup columns (Qiagen) using the manufacturer's guidelines for gel extraction, except for longer pre-elution column drying step (4 minutes at 10,000 g with open columns). Final DNA elution was done with 14 μL of MilliQ water. The purified DNA was quantified by Nanodrop 2000c spectro-photometer (Isogen Life Science) and utilized as DNA template for the upcoming evolutionary round.

## Quantitative PCR

Ten microliter reactions consisted of PowerUP SYBR Green Master Mix (Applied Biosystems), 400 nM each primer targeting the *p2* gene (976/977 ChD), and 1 μL of 100-fold diluted sample. The thermal cycling and data collection were performed on Quantstudio 5 Real-Time PCR instrument (Thermo Fisher), using the thermal cycling protocol 2 min at 50 °C, 5 min at 94 °C, 45 cycles of (15 sec at 94 °C, 15 s at 56 °C, 30 s at 68 °C), 5 min at 68 °C, and a melting curve from 65 °C to 95 °C. The concentration of nucleic acids was calibrated using 10-fold serial dilutions of corresponding standard DNA templates ranging from 1 fM to 1 nM. The data was analysed using the Quantstudio 5 Software (Thermo Fisher). Measured DNA concentrations indicated in Figs. 2 and 4 correspond to the amount (number of moles) of DNA encapsulated in liposomes (not digested by DNase I) divided by the total reaction volume (volume inside plus outside liposomes). Since outside-of-liposomes DNA is digested by DNase I, the measured DNA concentration is lower (typically between 0.1 and 1 pM at time zero) than the input DNA concentration (10 pM). DNA amplification folds were calculated with DNA concentrations at the IVTTR reaction end-point (generally 16 h)/DNA concentrations at the starting point of incubation (0 h). Fitting kinetics and statistical tests were performed using Excel.

## NGS of evolutionary intermediates: Library preparation, sequencing, and data analysis

DNA was PCR-amplified from 100-fold diluted liposome suspensions of evolutionary rounds as follows. Reactions were set up in 200 μL volume, 300 nM each primer, 400 μM dNTP, 20 μL of the diluted liposome suspension, and 4 units of KOD Xtreme Hotstart DNA poly-merase in Xtreme buffer. Thermal cycling was performed as follows: 2 min at 94 °C for polymerase activation, and 25–30 cycles of (98 °C for 10 s, 65 C for 20 s, 68 °C for 1.5 min). The DNA was then purified using QIAquick PCR purification buffers (Qiagen) and RNeasy MinElute Cleanup columns (Qiagen) using the manufacturer's guidelines for QIAquick PCR purification, except for longer pre-elution column dry-ing step (4 min at 10,000 g with open columns), and elution with 30 μL MilliQ water in the final step. The purified DNA was then prepared for deep sequencing using the Illumina Truseq DNA PCR free library pre-paration kit and deep sequenced using the Novaseq 6000 platform 150 bp paired end sequencing at Macrogen-Europe B.V.

To analyse NGS data, we utilized Galaxy, a web-based open-source platform for big data analysis at Usegalaxy.org. Using Galaxy available packages, we performed the following analysis steps. We mapped the paired reads to the *mod-ori-p2p3* DNA sequence using the BWA soft-ware package[79,80] in BAM format using default options. Next, we used the MergeSamFiles tool to merge BAM datasets from different rounds of evolution into one set and marked duplicates to examine the aligned records for duplicate molecules. We then used the BamLeftAlign tool to realign indels in homopolymers and microsatellite repeats. We next applied the Filter tool to filter data on read mapping quality (≥20) and proper read pairing. We then utilized the FreeBayes tool, a bayesian

genetic variant detector[81,82] to map and quantify the misalignments. The expected mutation rate was set to 0.0001. The requirement of minimal fraction of observations was set to 0.01 (for retrieving a list of all variants above 1%) or 0.001 (for quantifying all mutations above 0.1%). The requirement for the minimal count of observations sup-porting an alternate allele was set to 10. The data was then converted from VCF to tab-delimited format using the VCFtoTab-delimited tool. Further analysis, such as quantification of mutations above specified thresholds, and corrections of semantical errors on frequency calcu-lations were performed in Excel or in Mathematica (Wolfram Research).

## Sanger sequencing of isolated clones

The *mod-ori-p2p3* DNA pool isolated from the R11 round of Int-WT(1) evolution was cloned into a pUC57 vector using Gibson assembly. Primers 1313 ChD and 1314 ChD were used to amplify vector backbone and primers 1315 ChD and 1316 ChD were used to amplify the *p2-p3* region of R11 *mod-ori-p2p3* DNA, thereby adding 30 bp homology flanks to complement with pUC57 vector PCR. Phusion polymerase was used for PCR with the following protocol: 98 °C for 30 sec, 30 cycles (98 °C for 10 s – 64 °C for 15 s – 72 °C for 3 min) and 10 min at 72 °C for final extension. PCR products were digested with DpnI and further purified with a PCR clean-up kit (Promega). The R11 *p2-p3* PCR product was then cloned into the pUC57 backbone with Gibson Assembly (50 ng pUC57 backbone and 50 ng *p2-p3* insert). After incu-bation, 2.5 μL from the Gibson assembly mix were transformed into *E.coli* DH5α cells via heat-shock, and plated on LB/Amp. Twelve single colonies were selected for plasmid isolation and sent for Sanger sequencing at Macrogen (EZ-seq, with 5 different primers per clone using the primers 365-ChD, 952-ChD, 953-ChD, 982-ChD, and 1068-ChD).

## Purification of DNA polymerases, TP, SSB, and DSB proteins

Wild-type Φ29 DNA polymerase was expressed and purified as described in ref. 83. Terminal Protein, was expressed and purified as described in[52]. Single-stranded DNA binding protein and DSB were expressed and purified as described in ref. 84 and 52, respectively. Phi29 DNA polymerase variants S79G, A80T and S79G/A80T were obtained with the QuickChange site-directed mutagenesis kit (Strata-gene), using as template for the mutagenic reactions the plasmid pJLPM, which contains the wild-type *p2* gene[85], and following the manufacturer's instructions. The presence of the desired mutations, as well as the absence of additional ones, was determined by sequencing the entire gene. All DNAP variants were expressed in *E.coli* BL21(DE3) cells and further purified essentially as described for the wild-type DNAP[83].

## Bulk DNA replication with purified protein variants in PURE background

Bulk replication-transcription reactions were carried out with a mod-ified PURE system that did not contain solution III (ribosome). Replication-only reactions were performed with a customized PURE solution II minus T7 RNA polymerase (GeneFrontier Corp.). A 20 μL reaction solution was assembled with 10 μL solution I, 1 μL solution II, 10–20 mM ammonium sulphate, 300 μM dNTPs, 3 ng/mL of purified DNA polymerase variant, 3 ng/mL purified terminal protein, 375 μg/mL SSB protein, 105 μg/mL purified DSB protein, and 2 nM of indicated template DNA. Reactions were incubated for 16 h in a thermal cycler (C1000 Touch, Biorad) at a temperature of 30 °C. Before and after incubation, 1 μL of sample was taken for DNA quantification with qPCR. For DNA sample analysis by gel electrophoresis, pre-ran reaction solutions were incubated with 1 μL RNase A (4 mg/mL RNASe A solu-tion, Promega), and 1 μL RNase One (10 units/μL RNase ONE Ribonu-clease solution, Promega) for 1 or 2 h at 30 °C. Solutions were then supplemented with 1.5 mL EDTA (100 mM), 1.5 mL SDS (1%), and 10 to

20 mg Proteinase K (Thermo Scientific). Samples were incubated at 50 °C for 4 h, and column-purified using the QIAquick PCR purification buffers (Qiagen) and RNeasy MinElute Cleanup columns (Qiagen) using the manufacturer's guidelines for QIAquick PCR purification, except for an additional pre-elution column drying step (7 minutes at 10,000 g with open columns), and 10–20 min column incubation with 14 mL of ultrapure water (Merck Milli-Q) as the eluant for the final step. A 7 mL fraction of the eluate was mixed with 3 mL of 6x purple gel loading dye (NEB) and loaded in 0.7–1% agarose gel with ethidium bromide, following which DNA was separated using an electrophoresis system (Bio-Rad). A Bench Top 1-kb DNA Ladder (Promega) was used to estimate the size of DNA.

### Bulk DNA replication with purified protein variants in replication buffer

A 20 μL replication-only reaction solution was assembled, consisting of 1x Φ29 replication buffer (50 mM Tris-HCl (pH 7.5), 10 MgCl$_2$, 5% glycerol, 1 mM DTT, 10 or 20 mM of NH$_4$SO$_4$) (NEB), 0.625 ng/μL of purified DNAP, 1.25 ng/μL of purified TP, 375 μg/mL purified SSB, 105 μg/mL purified DSB, 400 μM of dNTPs, and 2 nM of linear DNA template. When required, purified DNAP and TP protein stocks were diluted with a buffer containing 100 mM NaCl, 0.05% Tween-20, and 25 mM Tris-HCl, prior to their addition into the reaction solution. Samples were incubated for 16 h in a thermal cycler (C1000 Touch, Biorad) at a temperature of 30 °C. Before and after incubation, 1 μL of sample was taken for DNA quantification with qPCR. For gel electrophoresis analysis, sample treatment and preparation of agarose gels were as described above.

### Co-translational labelling and gel fluorescence imaging of expressed proteins

Standard 20 μL PURE*frex*2.0 (GeneFrontier Corp.) reaction solutions were assembled on ice (10 μL solution I, 1 μL solution II, 0.5 μL solution III, 0.6 units/μL of Superase·In RNase inhibitor (Ambion), and 1 nM of linear template DNA) and supplemented with 1 μL of BODIPY-Lys-tRNALys (FluoroTect GreenLys, Promega) to incorporate fluorescently labelled lysine residues in the synthesized proteins. Samples were incubated at 37 °C for 16 hours for protein expression, after which they were treated with 1 μL RNase A (4 mg/mL RNASe A solution, Promega), and 1 μL RNase One (10 units/μL RNase ONE Ribonuclease solution, Promega) for 1–4 h at 37 °C. Ten microliters of treated samples were mixed with 4x Laemmli Sample buffer and DTT to reach a 15 μL volume and a final concentration of 1x Laemmli Sample buffer and 10 mM DTT, and were denatured for 5 min at 95 °C. Samples were analysed on a freshly prepared 12% SDS polyacrylamide gel electrophoresis (PAGE) gel. The loaded SDS gels were run for 15 min at 110 V, followed by 45 min at 180 V. Fluorescence imaging of the translation products was performed with a fluorescence gel imager (Typhoon, Amersham Biosciences). After fluorescence detection, the gels were stained overnight with Coomassie Brilliant Blue, de-stained overnight, and imaged on a Bio-Rad ChemiDoc Imager.

### Semi-continuous replication and evolution in liposomes

Swelling solutions for in-liposome IVTTR were prepared using 50 pM *mod-ori-p2p3* DNA template, PURE*frex*2.0, and DNA replication substrates, as explained above for the intermittent evolution protocol, except that DSB was included at a concentration of 105 μg/mL. Feeding vesicles were produced with the same protocol, except that DNA was not added, aliquoted and stored at −80 °C directly after the last freezing step. For IVTTR, samples were incubated for 4 or 16 h at 30 °C. Before and after incubation, 2 μL were collected with a cut pipette tip and diluted 100x with MiliQ-water for DNA quantitation by qPCR and PCR DNA recovery. When indicated, 0.5 μL of DNase I (Promega) was added to the liposome suspension (5–10 μL) in order to digest the outer DNA, so that only the DNA present inside liposomes was

quantified. To allow DNase to act, the IVTTR solution was incubated for 20 min at 30 °C, followed by 15 minutes incubation at 75 °C for DNAse heat inactivation. DNA recovery by PCR and gel electrophoresis analysis were performed as indicated above for the intermittent evolution protocol. To start a next evolution round, IVTTR-liposome samples were diluted either 100x or 10x as indicated with the feeding vesicle solution. The 100x dilution was realized in a two-step 10x dilution starting from 3 μL of IVTTR-liposome solution mixed with 27 μL of feeding vesicles. We gently pipetted up and down with a cut tip and kept a 2 μL sample to quantify the DNA concentration after feeding. Upon sample replenishment, 2 μL were again collected for qPCR DNA quantification. The remaining liposome solution was centrifuged for 5 minutes at 16000 r.c.f. at 4 °C. The tube was then dipped into liquid nitrogen for 5 seconds and left to thaw on ice for 10 min. Finally, liposomes were gently resuspended with a cut pipette tip, and incubated at 30 °C for a new IVTTR cycle. The procedure was repeated for a total of 4–8 cycles.

### Serial transfer of bulk IVTTR reactions

A 20 μL IVTTR reaction solution was assembled according to the protocol for preparation of the swelling solution for semi-continuous replication and evolution in liposomes and in the presence of 105 μg/mL DSB. The sample was incubated for 16 h at 30 °C. A 2 μL sample was taken before and after the incubation step for qPCR and PCR DNA recovery. DNA recovery and gel electrophoresis analysis were performed as indicated above for the semi-continuous evolution protocol. After incubation, 2 μL of the IVTTR solution was diluted 10x or 100x as indicated with a feeding solution of the same composition except that DNA was omitted. After gentle pipetting up and down, the next IVTTR round was started by incubating at 30 °C for 16 h. The procedure was repeated for a total of 6 rounds.

### LC-MS protein quantification

LC-MS/MS analysis was employed for the relative quantification of de novo synthesized DNAP and TP in bulk PURE reactions. 9-10 μL of the pre-ran PURE reaction (no older than one week, stored at −20 °C) was mixed with 3 μL of heavy isotope-labelled QconCAT($^{15}$N)[11], (peak quality was not good enough for quantitation), stored in a 50 mM Tris (pH 8.0) buffer containing 1 mM CaCl$_2$, and with 12–13 μL of freshly prepared digestion buffer (12.5 mM Tris-base, 12.5 mM Tris-HCl, 1 mM CaCl$_2$, 5 mM TCEP). Next, the mixture was vortexed vigorously, supplemented with 3.6 μL of 50 mM iodocamide, and incubated in the dark for 15 min. Then, 10 μL of trypsin (0.2 μg/μL) were added to each sample and the mixture was incubated overnight at 37 °C for protein digestion. The trypsin-digested samples were centrifuged at maximum speed (∼14,000–16,000 g) for 30 min. Fifteen microliters of the supernatant were collected and supplemented with 5–6 μL of 0.2% formic acid. The pH was checked with a pH strip to confirm the acidification of the solution (pH ∼2–4). The mixture was then transferred to a glass vial with a small insert for LC-MS/MS analysis. Measurements were performed on a 6460 Triple Quad LCMS system (Agilent Technologies, USA). About 5.5 μL of sample were injected per run into an ACQUITY UPLC Peptide CSH C18 Column (Waters Corporation, USA). The peptides were separated in a gradient of buffer A (25 mM formic acid in Milli-Q water) and buffer B (50 mM formic acid in acetonitrile) at a flow rate of 500 μL per minute and at a column temperature of 40 °C. The column was initially equilibrated with 98% buffer A. After sample injection, buffer A gradient was changed to 70% (over the first 20 min), 60% (over the next 4 min), and 20% (over the next 30 sec). This final ratio was maintained for another 30 sec and the column was finally flushed with 98% buffer A to equilibrate it for the next run. The selected peptides and their transitions for both synthesized proteins and heavy isotope-labelled QconCATs were measured by multiple reaction monitoring (MRM). The recorded LC-MS/MS data was analysed with Skyline for fraction calculation between unlabelled peptides

from DNAP and TP proteins. MS/MS measurement details for each of the analysed proteins can be found in Supplementary Table 4.

## Flow cytometry for liposome fusion assays

Liposomes were produced from lipid-coated beads prepared as explained above using 0.5 mol% of either Texas Red or Oregon Green membrane dyes. Swelling solutions consisted either of PURE*frex*2.0 or an mCherry-encoding DNA (2 nM) in homemade PURE buffer (PB) consisting of 20 mM HEPES-KOH, pH 7.6, 180 mM potassium glutamate, and 14 mM magnesium acetate. Vesicle fusion by F/T was achieved by mixing equivalent amounts (either 5 μL or 10 μL) of two different liposome populations, centrifuging for 1.5 minutes at 16,000 r.c.f, flash-freezing the sample tube in liquid nitrogen, and thawing on ice. For assaying liposome content mixing, samples were incubated for 3–6 h at 37 °C to allow for the expression of mCherry. One microliter of liposome samples was taken before and after F/T, diluted in 149 μL of swelling buffer, and filtered in 5 mL Falcon tubes (BD Falcon) with cell-strainer caps. Filtered diluted samples were pipetted into 96 U-shaped wells for flow cytometry analysis on a FACS Celesta flow cytometer (BD Biosciences). Liposomes were screened using the 488-nm laser line with 530/30 filter for detection of Oregon green, and the 561-nm laser line with 610/20 filter for detection of mCherry and Texas Red. Photon multiplier tube voltages were manually adjusted between 370 and 500 V for both laser lines, 375 V for the forward scatter light, and 260 V for the side scatter light. Loader settings were set to 50 μL injection volume with no mixing and 800 μL wash between sample runs. For each sample ~20000 events were recorded. The raw flow cytometry data was analysed and pre-processed to filter out possible aggregates and liposome debris using Cytobank (https://community.cytobank.org/), first by selecting the main population in the side- and forward-scattered light channels, then by filtering out low-fluorescence events[53].

## DNA templates, substrates and nucleotides used in Supplementary Fig. 13

Unlabelled nucleotides were purchased from GE Healthcare. The [γ-³²P]ATP (3,000 Ci/mmol) and [α-³²P]dATP (3000 Ci/mmol) were supplied by PerkinElmer. Oligonucleotides sp1 (5′-GATCA-CAGTGAGTAC), sp1c + 6 (5′-TCTATTGTACTCACTGTGATC), and M13 Universal Primer (5′-GTAAAACGACGGCCAGT) were purchased from Sigma-Aldrich. T4 polynucleotide kinase (T4PNK) was purchased from New England Biolabs. Oligonucleotide sp1 was 5′-labelled with ³²P using [γ-³²P]ATP (10 μCi) and T4PNK and further hybridised to oligonucleotide sp1c + 6 (1:2 ratio) to get the primer/template substrate sp1/sp1c + 6 for the Exonuclease/Polymerisation balance assays (see below). Oligonucleotides were annealed in the presence of 50 mM Tris-HCl (pH 7.5) and 0.2 M NaCl, heating to 90 °C for 10 min before slowly cooling to room temperature overnight. M13mp18 (+) strand ssDNA (Sigma-Aldrich) was hybridized to the universal primer as described above, and the resulting molecule was used as a primer/template complex to analyse processive DNA polymerisation coupled to strand displacement by the wild-type and variants of Φ29 DNAP. Terminal protein-Φ29 DNAP complex (TP-DNA) was obtained as described in ref. 86.

## Primed M13 DNA replication assay

The incubation mixture contained, in 25 μL, 50 mM Tris−HCl, pH 7.5, 10 mM MgCl₂, 1 mM DTT, 4% (v/v) glycerol, 0.1 mg/mL of BSA, 40 μM dNTPs and [α-³²P]dATP (1 μCi), 4.2 nM of primed M13mp18 ssDNA, and 60 nM of either the wild-type or the indicated mutant Φ29 DNA polymerase. After incubation at 30 °C for the indicated times, the reactions were stopped by adding 10 mM EDTA-0.1% SDS and the samples were filtered through Sephadex G-50 spin columns. For size analyses of the synthesised DNA, the labelled DNA was denatured by treatment with 0.7 M NaOH and subjected to electrophoresis in alkaline 0.7% agarose gels, as described in ref. 87. After electrophoresis the gels were dried and autoradiographed.

## TP-DNA amplification assay

The assay was performed essentially as described in ref. 88. The reaction mixture contained in a final volume of 25 μL, 50 mM Tris−HCl, pH 7.5, 10 mM MgCl₂, 20 mM ammonium sulphate, 1 mM DTT, 4% (v/v) glycerol, 0.1 mg/mL BSA, 80 mM of each dNTP and [α-³²P]dATP (1 μCi), 15 pM of TP-DNA, 3 nM of either wild-type or the indicated DNA polymerase variant, 6 nM of TP, 30 μM of SSB and 30 μM of DSB. After incubation for the indicated times at 30 °C, samples were processed as described for the TP-DNA replication assay and subjected to electrophoresis in alkaline 0.7% agarose gels, as described[87]. After electrophoresis, the gels were dried and autoradiographed.

## Exonuclease/polymerase balance assay

In a final volume of 12.5 μL, the incubation mixture contained 50 mM Tris-HCl, pH 7.5, 10 mM MgCl₂, 1 mM DTT, 4% (v/v) glycerol, 0.1 mg/mL BSA, 1 nM 5′-labelled sp1/sp1c + 6 substrate (a primer/template structure that contains a 6-nt 5′-protruding end, and therefore can be used as substrate for DNA-dependent DNA polymerisation and also for the exonuclease activity), 30 nM wild-type or mutant Φ29 DNA polymerase, and the indicated increasing concentrations of the four dNTPs (0–150 nM). After incubation for 5 min at 25 °C, the reaction was stopped by adding EDTA up to a final concentration of 10 mM. Reaction products were resolved by electrophoresis in 7 M urea-20% polyacrylamide gels and autoradiography. Polymerisation or 3′−5′ exonucleolysis was detected as an increase or decrease, respectively, in the size (15-mer) of the 5′-labelled primer.

## Electrophoretic mobility shift assay (EMSA)

The incubation mixture contained, in a final volume of 20 μL, 50 mM Tris-HCl, pH 7.5, 20 mM ammonium sulphate, 0.1 mg/mL BSA, 0.7 nM 5′-labelled sp1/sp1c + 6 primer/template hybrid, and the indicated amount of wild-type or mutant DNA polymerase. After incubation for 5 min at 4 °C, the samples were subjected to electrophoresis in pre-cooled 4% (w/v) polyacrylamide gels [80:1 acrylamide/bis-acrylamide (w/w)] containing 12 mM Tris acetate (pH 7.5) and 1 mM EDTA, and run at 4 °C in the same buffer at 8 V/cm[89]. After autoradiography, a stable interaction between the enzyme and the DNA was detected as a shift (retardation) in the migrating position of the labelled DNA.

## Statistics and reproducibility

Statistical tests (unpaired t-test, two-tailed *P* value) shown in Fig. 2c and Fig. 5b,c were performed using GraphPad Prism. For data presented in graphs, the figure legends provide the number of biological replicates that were performed. For data presented in graphs, the figure legends provide the number of biological replicates that were performed. For gels presented in Figs. 2d, 3c,f, and in Supplementary Figs. 1a, 4, 8b, 9b,d,e and 19, the experiments were performed one time. Gels presented in Fig. 1b, Supplementary Fig. 1a,b are representative of at least two gels from independent experiments.

## Reporting summary

Further information on research design is available in the Nature Portfolio Reporting Summary linked to this article.

## Data availability

Data are available in the main manuscript, Supporting Information, and Supplementary Data 1. Protein mass spectrometry data are available on Panorama Public under ProteomeXchange ID PXD054024 and accession URL: https://panoramaweb.org/qxuWjQ.url. Raw NGS sequencing data have been uploaded to ENA (European Nucleotide Archive) under project_ID PRJEB75735. The previously published protein structure used in this work is available from the Protein Data Bank

under ID 2EX3. Raw data in multiple labelled files (Excel and GraphPad) are available within a zipped folder named 'Source Data'. Source data are provided with this paper.

## Code availability

Galaxy workflow for mapping and quantifying mutation frequencies is available on the GitHub repository (https://github.com/DanelonLab/Illumina-NGS-Mutation-Mapping) and zenodo, with https://doi.org/10.5281/zenodo.13757341.

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

## Acknowledgements

We would like to thank Ilja Westerlaken for single clone preparation and Sanger sequencing analysis, Elisa Godino for assisting with the characterization of some DNA variants, Anna de Jong for the flow cytometry experiments, and Laura Sierra Heras for performing a replicate of Bulk-WT. The research was funded by the NWO Gravitation programs "BaSyC –

Building a synthetic cell" (024.003.019) and "NanoFront – Frontiers of Nanoscience". CD acknowledges funding from ANR (ANR-22-CPJ2-0091-01). ZA acknowledges funding from the European Union's Horizon 2020 research and innovation programme under the Marie Skłodowska-Curie grant agreement no. 707404. ARS acknowledges funding from a NanoFront-Casimir Research School PhD grant. MdV acknowledges funding by MCIN/AEI/10.13039/501100011033 (Grant PID2020-115978GB-I00). We specifically acknowledge the contributions of author Andreea R. Stan, who sadly passed away and did not see the final version of the paper.

## Author contributions

CD conceived and supervised the project; ZA, AMRS, ARS, AC, and AdP designed, performed experiments, and performed data analysis; MdV designed experiments and performed data analysis; YR performed data analysis; ZA, AMRS and CD wrote the paper with input from all co-authors.

## Competing interests

The authors declare no competing interests.
