## [Transparent Peer Review file · Nature Communications]

Darwinian Evolution of Self-Replicating DNA in a Synthetic Protocell

Corresponding Author: Professor Christophe Danelon

Version 0:

Reviewer comments:

Reviewer #1

(Remarks to the Author)

The article entitled "Darwinian Evolution of Self-replicating DNA in a synthetic protocell" describes Darwinian evolutionary experiments in liposomes by using the phi29-based DNA replication system. In their system, TP and DNAP are expressed from the template linear DNA in a reconstituted transcription/translation system. The authors demonstrated a series of evolutionary experiments in liposomes in an intermitted or semi-continuous manner. In both the intermitted and semi-continuous serial dilution experiments, they showed that the DNA replication efficiencies improved within 10 rounds and found some mutations accumulated in both DNAP and TP genes. Among the mutations, they found that two mutations (S79G and A80T) are beneficial for DNA replication in liposomes. I think that the enormous experiments conducted here are sufficient for their conclusion and the findings are worth publishing. Especially, the observation of Darwinian evolutionary processes in liposomes are significant advance in the synthetic cell field. The manuscript is written concisely and I think can be published as it is after addressing the minor points listed below.

1. In the second paragraph of page 10, the authors wrote "This experiment,...was conducted in the presence of DSB..." In the former part, they wrote DSB can be omitted in in-lipo reaction. What is the difference?
2. In the first paragraph of the CONCLUSION AND OUTLOOK, the authors wrote that the evolution of their system is faster than the previous Qbeta system. However, I think the speed of evolution mainly depends on whether beneficial mutations exist in the library and how much the mutation enhances the fitness, not an issue related to the replication scheme. Actually, the first beneficial mutations even in the Qbeta case became dominated in 10 rounds (see Fig. 2 of Ichihashi et al.(2015) *Molecular Biology and Evolution*,32(12),3205-3214). Nevertheless, I agree with the authors on the point that this DNA-based system is more suitable for synthetic cell research because increasing gene numbers is very difficult for the Qbeta-based replication system.
3. Is the DNA concentration in liposome IVTTR (e.g., Fig. 4a) the expected concentration in liposome or the concentration after averaging with the outer solution? It would be more helpful if there is an explanation in the legend.

Reviewer #2

(Remarks to the Author)

Abil et al. demonstrated the evolution of an artificial DNA replication system within liposomes. The system contained a linear DNA template encoding two Phi29 bacteriophage-derived proteins including a DNA polymerase, and the DNA replicated using the self-encoded proteins that were expressed in a co-encapsulated cell-free translation system. Using this system, the authors performed evolution experiments in "intermittent (Int)" and "semi-continuous (Con)" schemes. The former scheme included steps of manual DNA amplification and re-encapsulation into liposomes, whereas the latter scheme lacked such manual interventions to support DNA amplification. The authors demonstrated continuous DNA replication up to certain rounds and the gradual accumulation of mutations in both experiments. They also identified a few mutations that likely supported adaptive evolution in intermittent evolution experiments. While there are similar systems capable of continuous genome replication and evolution based on artificial linear RNA genomes or circular DNA genomes, the linear DNA version

could be advantageous for developing a large evolving genome, a crucial element for building a synthetic cell. Thus, I found the research novel and important. However, I also found multiple concerns, especially over the demonstration of sustained replication and adaptation of DNA in the semi-continuous evolution experiment, a key experiment to support the conclusion, and I expect the authors to address the following comments before publication.

General comments:

The authors claimed that they demonstrated persistent DNA replication and adaptive evolution in a semi-continuous evolutionary setting (Con-WT, in addition to intermittent setting), i.e., “Darwinian evolution of self-replicating DNA” as in the title, shown in Fig. 3 and Table 1. I have several concerns about this point.

1. The Con-WT experiment was performed up to only round 5, while Bulk-WT lasted to round 6. Fig. 3c showed the appearance of a product shorter than the genome at round 4 of Con-WT, and the product seemed to be maintained at the next (final) round. If this band corresponds to a fast-replicating parasite, the target DNA replication may stop in the next few rounds. To demonstrate the persistency of the system, I believe that the authors should continue the experiment a little further (for example, up to round 10) to see whether the system is (apparently) fully persistent or semi-persistent.
2. I thought that the different persistency of Con-WT and Bulk-WT may have resulted from different dilution factors (100- and 10-fold dilutions for Con-WT and Bulk-WT, respectively). For example, even if a small number of parasitic DNA appeared, sufficient dilution (i.e., 100-fold dilution) may eliminate them before the next round. Different dilution factors could also affect the extent of replication (see also my comment 10), and low replication capacity would also limit the capacity of adaptive evolution. Have the authors done Bulk-WT with 100-fold dilution (or Con-WT with 10-fold dilution)?
3. The biochemical characterization of evolution was conducted only with variants obtained in intermittent evolution experiments. Therefore, whether adaptive evolution occurred in Con-WT was unclear, although the authors showed the accumulation of mutations that are indicative of such evolution. In addition, I was a little concerned about the (much) faster accumulation of mutations in Con-WT compared to Int-WT, even though the expected mutation rate was lower for Con-WT. If this represents enrichment during evolution based on increased fitness (e.g., the ability to replicate), the dominant DNA (with some or all of the five mutations) is expected to propagate highly efficiently. I suggest the authors confirm some selective advantages of a dominant variant.

Following are other comments, roughly in the same order as they appear in the manuscript:

4. (p5) I do not think “vsv-r1” and “vsv-r2” are names common enough to use without describing what they stand for. Please specify them.
5. (p5, Fig. 1b) The authors wrote, “During bulk IVTTR, a main self-replication product of size 3.2 kb was generated, as well as an unexpected additional band of size 1.4 kb (Fig. 1b),” but a similar amount of 1.4 kb band appeared without dNTP (i.e., without DNA replication), which I thought may indicate their presence before IVTTR. I ask the authors to describe more about the appearance of this band. Also, the caption of Fig. 1b states it was not bulk IVTTR but in-liposome IVTTR.
6. Relatedly, I could not (confidently) find methods for in-liposome IVTTR in the Materials and Methods section. It may be the same step as a part of “Intermittent evolution of self-replicating DNA,” but I was unsure without further guidance. Please clarify where to see when one would like to know the method of in-liposome IVTTR.
7. (p5) The authors wrote, “The leader was designed to form a hairpin RNA structure (Fig. 1c), which was found to be important for gene expression in this system (Supplementary Note 1).” However, I could not find a part to investigate the importance of hairpin structures in Supplementary Note 1. Unless the authors examine leader sequences that do not form hairpin structures, the importance of hairpin structure is difficult to claim.
8. (p7) “At each round of” intermittent evolution, the authors “encapsulated in liposomes the IVTTR reaction mix (no DSB added) along with 10 pM DNA,” but it largely differs from concentrations shown in plots of Figs. 2b, e, f (especially Int-WT (2)). What did the authors measure here? Is the discrepancy because most DNA was not encapsulated and digested by DNase I (and thus apparently diluted)?
9. (p10) The authors claimed that they conducted Con-WT “in the presence of DSB due to rapid extinction of the replicator in the absence of DSB in the tested conditions (Fig. S6c-f),” although the absence of DSB did not diminish DNA replication at the initial round. However, the authors applied only 2-fold dilution instead of 100-fold as in Fig. 3, and I thought that the reaction may have become inefficient after round 1 due to poor supplement of fresh nutrients and translation systems. Or, did the authors try to convey that the DNA replication outside liposomes (Fig. S6f) is a key for sustained replication?
10. (p10) Fig. 3g was not mentioned in the text, and thus I am not sure what the authors tried to show. However, without any explanation, this panel could cause misinterpretation due to different dilution factors applied to Con- and Bulk-WT. There was generally a higher capacity for replication in Con-WT than in Bulk-WT, and fold-amplification does not necessarily relate to evolution, as opposed to intermittent evolution, where the authors tried to keep the initial DNA concentration constant at each round.
11. (p11) The authors wrote, “the abundance of p2 in the bulk evolution experiment decreased faster than the other regions (Fig. 3i),” but it is unclear to my eye. The dynamics of p2 were different clearly from those of ori-L but appeared similar to

those of p3 (and ori-R) despite different amplification at the initial round.

12. (p12) The authors “mapped and extracted the frequency of occurrence of all the point mutations that were detected at a frequency of at least 1% in at least one of the evolutionary rounds (Table 1, Fig. S8).” However, according to the captions of both Table 1 and Fig. S8, they appeared to show mutations detected at frequencies of at least 5%. Which are correct?

13. (p13) The authors found possible contamination of DNA variants from the Int-WT experiment into the Int-Mut experiment, which may affect the results shown in Fig. 2. I believe that this information should be noted in the “Evolution of self-replicators over multiple rounds of intermittent evolution” section (or p8) to avoid any misinterpretations by readers (especially those who do not read after).

14. (p14) In Fig. S10, it is hard to distinguish the 24-bp swapping from other point mutations. Please consider changing the color to highlight the swapping.

15. (p14) Do the authors have any ideas about how variants of “a large deletion of around 200 bases, preserving the ORF but removing most of the right origin” in the Int-WT experiment could replicate without the replication origin?

16. (p15) The authors suggested using a mutator DNAP variant to accelerate evolution in semi-continuous evolution experiments. Why did not the authors use it in the first place, while using it for intermittent evolution experiments? I will not ask the authors to conduct a new experiment, but if it has already been done (e.g., performed but the population went extinct), please disclose the data.

17. (p15, Fig. 4) Please specify what statistical test was used to determine the p-values.

18. (p15) The authors stated that their finding of Fig. 4d “suggests a mutational fitness advantage in response to compartmentalization in liposomes,” but I do not think this is necessarily the case. For example, the variant may show a higher initial replication rate as in panel c. Was not the replication already saturated?

19. (p16) The authors investigated the activity of purified polymerases that harbor the enriched mutations in bulk IVTT reactions (Fig. S15) and concluded that “it is unlikely that the amino acid residue substitution in the translated DNAP protein improves its replication activity.” However, because the enhanced IVTTR was so far observed only in in-liposome conditions (Fig. 4), it is conceivable that the purified polymerase variants may act better in liposomes.

20. (p17) The authors stated, “the discovery that adaptive evolution arose relatively fast (within 10 rounds of evolution)... faster than with the RNA/Q β -replicase system 38,42 (within 100 rounds)...” I think that the “discovery” was referred to as the intermittent evolution experiments. However, in intermittent evolution experiments using an RNA/Q β -replicase system, adaptive evolution could arise just as fast. For example, Mizuuchi et al., ACS Synth. Biol., 2015 (ref 40) demonstrated such evolution within 11 rounds.

Version 1:

Reviewer comments:

Reviewer #2

(Remarks to the Author)

I appreciate the authors' efforts in revising the manuscript, including performing additional experiments, which I believe have significantly improved the research. Most of my previous concerns have been resolved. I now have only a few relatively minor comments, as follows:

1. (Regarding the original comment/answer 2) The new results support evidence of “persistent” DNA replication in liposomes. My only remaining concern is about the different extents of DNA replication observed in the two Bulk-WT experiments (Figs. 3e and S9a). I initially thought that persistent DNA replication may occur in Bulk-WT if a 100-fold dilution is applied because the amplification in the initial round was >100-fold. However, in the new experiment shown in Fig S9, the replication at the initial round was only <5-fold, despite the same initial conditions for both experiments (please correct me if I am wrong). Thus, the decreasing trajectory of DNA concentration and amplification in the new experiment may have resulted from (somehow) significantly different efficiencies of IVTTR. Could the authors provide an explanation for this?

2. (Regarding the original comment/answer 3) I apologize for my confusing comment about the “faster accumulation” of mutations in Con-WT. I probably meant to write “faster enrichment,” but in any case, now I agree with the authors that this could be explained within the context of Darwinian evolution.

3. (Regarding the original comment/answer 7) In the new Fig. S1a, the incubation time for the far right lane is missing.

4. (Regarding the original comment/answer 8) I think that “Figs 2 and 3” in the revised manuscript were meant to be Figs 2 and 4.

Subject: Revision for Manuscript **NCOMMS-24-26213**

Title: "Darwinian Evolution of Self-Replicating DNA in a Synthetic Protocell"

Author(s): Zhanar Abil, Ana María Restrepo Sierra, Andreea R. Stan, Amélie Chêne, Alicia del Prado, Miguel de Vega, Yannick Rondelez & Christophe Danelon

We are delighted by the positive evaluation and are grateful to the two referees for their constructive comments that helped us improve the manuscript. Please, find below our point-by-point response (in *blue* text) to the reviewer comments (in black text). Modifications in the manuscript or supplementary information are highlighted in *red* text.

Reviewer comments:

Reviewer #1 (Remarks to the Author):

The article entitled "Darwinian Evolution of Self-replicating DNA in a synthetic protocell" describes Darwinian evolutionary experiments in liposomes by using the phi29-based DNA replication system. In their system, TP and DNAP are expressed from the template linear DNA in a reconstituted transcription/translation system. The authors demonstrated a series of evolutionary experiments in liposomes in an intermitted or semi-continuous manner. In both the intermitted and semi-continuous serial dilution experiments, they showed that the DNA replication efficiencies improved within 10 rounds and found some mutations accumulated in both DNAP and TP genes. Among the mutations, they found that two mutations (S79G and A80T) are beneficial for DNA replication in liposomes. I think that the enormous experiments conducted here are sufficient for their conclusion and the findings are worth publishing. Especially, the observation of Darwinian evolutionary processes in liposomes are significant advance in the synthetic cell field. The manuscript is written concisely and I think can be published as it is after addressing the minor points listed below.

1. In the second paragraph of page 10, the authors wrote "This experiment,...was conducted in the presence of DSB..." In the former part, they wrote DSB can be omitted in in-lipo reaction. What is the difference?

Answer: We conducted the Int-WT evolution experiments in the absence of DSB since we found that it is dispensable in liposomes (Fig. S2). In Con-WT, we omitted DNase I outside of liposomes because it could be taken up inside liposomes during F/T and inhibit in-liposome replication in the next round. To maintain the complete pool of DNA variants from one round to the next, we chose to enable outside-of-liposome DNA replication by adding DSB. This way, the DNA that replicated in the interior of liposomes but leaked out during F/T (about 50%, see Fig. S6a,b) can potentially be amplified and re-encapsulated in a subsequent round, hence propagate, maintaining the pool of variants that would otherwise be washed out by dilution if DNA replication outside liposomes was disabled (no DSB).

We clarified in the revised manuscript, on page 10, why DSB was added in the semi-continuous experiments: "This experiment, which we dubbed Con-WT, was conducted in the presence of DSB to maintain the complete pool of replicating DNA variants by enabling outside-of-liposome IVTTR. DNA that replicated in the interior of liposomes but leaked out during F/T (about 50%, see Fig. S6a,b) can potentially be amplified and re-encapsulated in a subsequent round, hence propagate, maintaining the pool of variants that would otherwise be washed out by dilution if DNA replication outside liposomes was disabled (no DSB)."

2. In the first paragraph of the CONCLUSION AND OUTLOOK, the authors wrote that the evolution of their system is faster than the previous Qbeta system. However, I think the speed of evolution mainly depends on whether beneficial mutations exist in the library and how much the mutation enhances the fitness, not an issue related to the replication scheme. Actually, the first beneficial mutations even in the Qbeta case became dominated in 10 rounds (see Fig. 2 of Ichihashi et al.(2015) Molecular Biology and Evolution,32(12),3205-3214). Nevertheless, I agree with the authors on the point that this DNA-based system is more suitable for synthetic cell research because increasing gene numbers is very difficult for the Qbeta-based replication system.

Answer: We thank the reviewer for raising this point. Indeed, Mizuuchi et al. (ref. 40) demonstrated a fast adaptive evolution of the Q β RNA replicase system, which was already evident within 10 rounds of evolution. We concede that our initial claim was not delivered correctly. In the updated manuscript, on page 17, we deleted the part “, and faster than with the RNA/Q β -replicase system^{38,42} (within 100 rounds)”. The sentence now reads: “The discovery that adaptive evolution arose relatively fast (within 10 rounds of evolution) compared to repetitive, concatemeric DNA replication⁴⁵, where mutation effects average out due to multiple gene copies per molecule, makes our DNA self-replicating mechanism a good candidate for implementation in an evolving synthetic cell.”

3. Is the DNA concentration in liposome IVTTR (e.g., Fig. 4a) the expected concentration in liposome or the concentration after averaging with the outer solution? It would be more helpful if there is an explanation in the legend.

Answer: The DNA concentration indicated in Figs. 2b,c,e,f, 4a corresponds to the amount (number of moles) of DNA encapsulated in liposomes (not digested by DNase I) divided by the total reaction volume (volume inside plus outside liposomes). Since DNA outside liposomes is digested by DNase I, the measured DNA concentration is lower (typically between 0.1 and 1 pM at time zero) than the input DNA concentration (10 pM). We clarified this point on pages 21-22 in the Methods section “Quantitative PCR”: “Measured DNA concentrations indicated in Figs. 2 and 3 correspond to the amount (number of moles) of DNA encapsulated in liposomes (not digested by DNase I) divided by the total reaction volume (volume inside plus outside liposomes). Since outside-of-liposomes DNA is digested by DNase I, the measured DNA concentration is lower (typically between 0.1 and 1 pM at time zero) than the input DNA concentration (10 pM).”

Reviewer #2 (Remarks to the Author):

Abil et al. demonstrated the evolution of an artificial DNA replication system within liposomes. The system contained a linear DNA template encoding two Phi29 bacteriophage-derived proteins including a DNA polymerase, and the DNA replicated using the self-encoded proteins that were expressed in a co-encapsulated cell-free translation system. Using this system, the authors performed evolution experiments in “intermittent (Int)” and “semi-continuous (Con)” schemes. The former scheme included steps of manual DNA amplification and re-encapsulation into liposomes, whereas the latter scheme lacked such manual interventions to support DNA amplification. The authors demonstrated continuous DNA replication up to certain rounds and the gradual accumulation of mutations in both experiments. They also identified a few mutations that likely supported adaptive evolution in intermittent evolution experiments. While there are similar systems capable of continuous genome replication and evolution based on artificial linear RNA genomes or circular DNA genomes, the linear DNA version could be advantageous for developing a large evolving genome, a crucial element for building a synthetic cell. Thus, I found the research novel and important. However, I also found multiple concerns, especially over the demonstration of sustained replication and adaptation of DNA in the semi-continuous evolution experiment, a key experiment to support the conclusion, and I expect the authors to address the following comments before publication.

Answer: We are grateful to the reviewer for the careful reading, detailed comments, and for finding our study novel and important.

General comments:

The authors claimed that they demonstrated persistent DNA replication and adaptive evolution in a semi-continuous evolutionary setting (Con-WT, in addition to intermittent setting), i.e., “Darwinian evolution of self-replicating DNA” as in the title, shown in Fig. 3 and Table 1. I have several concerns about this point.

1. The Con-WT experiment was performed up to only round 5, while Bulk-WT lasted to round 6. Fig. 3c showed the appearance of a product shorter than the genome at round 4 of Con-WT, and the product seemed to be maintained at the next (final) round. If this band corresponds to a fast-replicating parasite, the target DNA replication may stop in the next few rounds. To demonstrate the persistency of the system, I believe that the authors should continue the experiment a little further (for example, up to round 10) to see whether the system is (apparently) fully persistent or semi-persistent.

Answer: It is always possible that the self-replicator will get extinct at round $R + 1$. The appearance of a shorter product at round 4 might be indicative of the emergence of parasite that will eventually take over the full-length replicator. With 'persistent' we mean that parasites have not taken over the self-replicator yet. We think that our data (including the new ones reported in **New Figs. 8 and 9**, see below) demonstrate that the presence of liposomes slows down the extinction of self-replication compared to bulk reactions, which we refer to as "persistent survival" or "persistent self-amplification". For clarity, and to avoid speculating on whether the system is fully or semi-persistent, we define on page 11 the persistency of the system as "the self-replicator has not been washed away or outcompeted by short replicons."

2. I thought that the different persistency of Con-WT and Bulk-WT may have resulted from different dilution factors (100- and 10-fold dilutions for Con-WT and Bulk-WT, respectively). For example, even if a small number of parasitic DNA appeared, sufficient dilution (i.e., 100-fold dilution) may eliminate them before the next round. Different dilution factors could also affect the extent of replication (see also my comment 10), and low replication capacity would also limit the capacity of adaptive evolution. Have the authors done Bulk-WT with 100-fold dilution (or Con-WT with 10-fold dilution)?

Answer: We performed an independent continuous evolution experiment, this time by reducing the dilution factor to 10-fold. To limit replication of external DNA, the IVTTR incubation time was reduced from 16 to 4 hours. The results are summarised in the figure below (**New Fig. S8**). Total (inside and outside of liposomes) DNA concentration did not noticeably change in the first round of IVTTR (**New Fig. S8a**), which is likely due to external DNA amplification kinetics being not high enough to reach a log phase within 4 hours (**Fig. S6g**), and internal DNA amplification being unnoticeable as the entrapped DNA represents only a small fraction of the total DNA pool. During the three following rounds, total DNA concentration remained relatively constant due to roughly equal DNA amplification and dilution rates. Finally, DNA concentration gradually increased 1700-fold from round 4 to round 8, corresponding to 225-fold amplification at round 8 alone (**New Fig. S8a,c**). Analysis of DNA species flanked with origins of replication revealed retention of the full-length replicator but also accumulation of lower-sized products that appeared at round 6 (**New Fig. S8b**). Quantitative PCR targeting multiple regions scanning the entire length of *mod-ori-p2p3* was also carried out. The *p2* gene driving replication follows the dynamic pattern of the other targeted regions (**New Fig. S8d**), corroborating the observations in Con-WT (16 h) (main text **Fig. 3h**).

We conclude that the replicator evolution in the presence of liposomes displays a slower accumulation of parasites compared to Bulk-WT with the same dilution factor ($\times 10$). The capacity of adaptive evolution cannot directly be assessed because the gradual increase of initial DNA concentration in the course of the experiment may affect the extent of replication.

New Fig. S8: Con-WT with 10-fold dilution. **a)** Trajectories of *mod-ori-p2p3* concentrations in liposomes in the continuous evolutionary campaign, Con-WT(4 h), as measured by qPCR (*p2* gene). The IVTTR was incubated for 4 hours and liposomes were diluted 10-fold between rounds. **b)** Size analysis of PCR-amplified DNA during Con-WT(4 h) by agarose gel electrophoresis. The arrowhead indicates the full-length replicator. **c)** Amplification of *mod-ori-p2p3* during the Bulk-WT, Con-WT (16 h) and the new Con-WT (4 h) experiments. **d)** DNA quantification of the different targeted regions. Colour coding is the same as in main text Fig. 3h,i (*p2* gene is in orange).

Furthermore, we followed the reviewer's suggestion and performed a Bulk-WT with 100-fold dilution. The trajectory of *mod-ori-p2p3* and the amplification for each round are displayed below:

New Fig. S9: Bulk-WT with 100-fold dilution. **a)** Trajectories of *mod-ori-p2p3* concentrations and amplification folds as measured by qPCR (*p2* gene). **b)** Size analysis of PCR-amplified DNA by agarose gel electrophoresis. The arrowhead indicates the full-length replicator.

As expected, the self-replicator rapidly gets extinct, as observed with qPCR (New Fig. S9a) and gel electrophoresis (New Fig. S9b); the amplification efficiency is not sufficient to compensate for the large dilution factor. These results contrast with the persistency of the full-length replicator in Con-WT (16 h) in which the same dilution factor was applied (main text Fig. 3b).

We added a new Supplementary Note 5 that summarizes the main results and we modified the manuscript on page 12 as: “To ascertain that the different persistency of Con-WT and Bulk-WT did not result from different dilution factors (100- and 10-fold dilutions for Con-WT and Bulk-WT, respectively), we repeated both experiments this time by changing the dilution factor (Supplementary Note 5). The results corroborate our finding that the DNA self-replicator persists longer in the presence of liposomes than in bulk reactions (Figs. S8 and S9).”

3. The biochemical characterization of evolution was conducted only with variants obtained in intermittent evolution experiments. Therefore, whether adaptive evolution occurred in Con-WT was unclear, although the authors showed the accumulation of mutations that are indicative of such evolution. In addition, I was a little concerned about the (much) faster accumulation of mutations in Con-WT compared to Int-WT, even though the expected mutation rate was lower for Con-WT. If this represents enrichment during evolution based on increased fitness (e.g., the ability to replicate), the dominant DNA (with some or all of the five mutations) is expected to propagate highly efficiently. I suggest the authors confirm some selective advantages of a dominant variant.

Answer: We agree with the reviewer that it is unclear from the data presented if adaptive evolution took place in the case of Con-WT. Therefore, we were careful not to claim this in the manuscript. It is clear from Table 1 that mutations accumulated and got enriched in the population, which could have been a result of a genetic drift or adaptation, both of which would support that Darwinian evolution took place.

On the other hand, the reviewer assumes that there was much faster accumulation of mutations in Con-WT. This is not so obvious to us from the analysis presented. A robust analysis of mutation accumulation rates would probably be useful in a follow-up study.

We agree that in this study, we only characterized a few mutations, and that a systematic characterization of all mutations could have provided a more complete picture of the genotype-phenotype landscape of the two genes, but this was a priori not the focus of this study.

Following are other comments, roughly in the same order as they appear in the manuscript:

4. (p5) I do not think “vsv-r1” and “vsv-r2” are names common enough to use without describing what they stand for. Please specify them.

Answer: On page 5, we wrote “and either vsv-r1 and vsv-r2 (from vesicular stomatitis virus (VSV) internal terminator) or a T7 transcription terminator”.

5. (p5, Fig. 1b) The authors wrote, “During bulk IVTTR, a main self-replication product of size 3.2 kb was generated, as well as an unexpected additional band of size 1.4 kb (Fig. 1b),” but a similar amount of 1.4 kb band appeared without dNTP (i.e., without DNA replication), which I thought may indicate their presence before IVTTR.

I ask the authors to describe more about the appearance of this band. Also, the caption of Fig. 1b states it was not bulk IVTTR but in-liposome IVTTR.

Answer: We thank the reviewer for noticing the inconsistency. The gel-electrophoresis image in Fig. 1b is indeed in-liposome IVTTR followed by recovery of DNA by PCR, as indicated in the figure legend, but the manuscript text mistakenly stated that Fig. 1b referred to results in bulk IVTTR. In the updated manuscript, page 5, we now claim:

“During bulk (Fig. S1b) and in liposome (Fig. 1b) IVTTR, a main self-replication product of size 3.2 kb was generated, as well as an unexpected additional band of size 1.4 kb.”

The 1.4 kb band without dNTP in Fig. 1b is the result of the PCR step that we use to recover DNA from in-liposome IVTTR (Fig. 1b legend). This band does not appear in bulk IVTTR in the absence of dNTPs (Fig. S1b).

6. Relatedly, I could not (confidently) find methods for in-liposome IVTTR in the Materials and Methods section. It may be the same step as a part of “Intermittent evolution of self-replicating DNA,” but I was unsure without further guidance. Please clarify where to see when one would like to know the method of in-liposome IVTTR.

Answer: For clarity, we renamed the intermittent evolution section as “*Intermittent evolution: in-liposome IVTTR.*”. Moreover, we now more clearly separate the part on the preparation of in-liposome IVTTR reactions and that on the DNA recovery strategy for the Int-WT evolutionary campaign.

7. (p5) The authors wrote, “The leader was designed to form a hairpin RNA structure (Fig. 1c), which was found to be important for gene expression in this system (Supplementary Note 1).” However, I could not find a part to investigate the importance of hairpin structures in Supplementary Note 1. Unless the authors examine leader sequences that do not form hairpin structures, the importance of hairpin structure is difficult to claim.

Answer: Indeed, we also investigated modified ori-p2p3 templates that transcribe an RNA of the same length but that lack the leader hairpin structure upstream of the TP gene. One of these templates, mod-ori-p2p3-wo/hp, was originally part of the Fig. S1a SDS-PAGE gel, which we later cropped for simplicity of data presentation. In response to your question, we restored the original SDS-PAGE gel with all the original lanes present (please see below) in Fig. S1a. Predicted RNA fold structure from this additional template is shown in the right panel below (Also included in Fig. S1a). Mod-ori-p2p3-wo/hp does not self-amplify (data not shown) and expresses the TP protein poorly (Fig. S1a).

New Fig. S1a

We updated the manuscript text in Supplementary Note 1 as follows:

“We therefore constructed modified ori-p2p3 templates wherein 33 bp of the leader sequence upstream of the p3 gene was replaced by an alternative sequence that, when transcribed, was predicted to either not form (template mod-ori-p2p3-wo/hp) or form (template mod-ori-p2p3) a similar mRNA stem loop. Neither of these alternative leader sequences have any sequence similarity to the original leader sequence. Only the mod-ori-p2p3 template demonstrated a TP expression level similar to the original ori-p2p3 (Fig. S1a), whereas the mod-ori-p2p3-wo/hp expressed TP poorly (Fig. S1a). These data suggest that the hairpin structure in the leader sequence of transcribed RNA is important for translation in the PURE system. Mod-ori-p2p3 demonstrated self-replication ability similar to the original ori-p2p3 (Fig. 1b, Fig. S1b), although it seems to produce additional smaller DNA products of sizes roughly 1.1 and 2.5 kb.”

8. (p7) “At each round of” intermittent evolution, the authors “encapsulated in liposomes the IVTTR reaction mix (no DSB added) along with 10 pM DNA,” but it largely differs from concentrations shown in plots of Figs. 2b, e, f (especially Int-WT (2)). What did the authors measure here? Is the discrepancy because most DNA was not encapsulated and digested by DNase I (and thus apparently diluted)?

Answer: *The reviewer is correct. The indicated DNA concentration of 10 pM corresponds to the input DNA concentration before compartmentalization and digestion of outside-of-liposome DNA by DNase I, whereas the DNA concentration indicated in Figs. 2b,c,e,f, 4a corresponds to the amount (number of moles) of DNA encapsulated in liposomes (not digested by DNase I) divided by the total reaction volume (volume inside plus outside liposomes). Since DNA outside liposomes is digested by DNase I, the measured DNA concentration is lower (typically between 0.1 and 1 pM at time zero) than the input DNA concentration (10 pM). We clarified this point on pages 21-22 in the Methods section “Quantitative PCR”: “Measured DNA concentrations indicated in Figs. 2 and 3 correspond to the amount (number of moles) of DNA encapsulated in liposomes (not digested by DNase I) divided by the total reaction volume (volume inside plus outside liposomes). Since outside-of-liposomes DNA is digested by DNase I, the measured DNA concentration is lower (typically between 0.1 and 1 pM at time zero) than the input DNA concentration (10 pM).”*

9. (p10) The authors claimed that they conducted Con-WT “in the presence of DSB due to rapid extinction of the replicator in the absence of DSB in the tested conditions (Fig. S6c-f),” although the absence of DSB did not diminish DNA replication at the initial round. However, the authors applied only 2-fold dilution instead of 100-fold as in Fig. 3, and I thought that the reaction may have become inefficient after round 1 due to poor supplement of fresh nutrients and translation systems. Or, did the authors try to convey that the DNA replication outside liposomes (Fig. S6f) is a key for sustained replication?

Answer: *In the experiment shown in Fig. S6c, the absence of DSB did not diminish DNA replication at the initial round because DNA replication occurred both inside and outside liposomes, and in vesiculo IVTTR is not hampered.*

We agree that 2-fold dilution may not be sufficient to ensure effective replenishment of fresh nutrients and translation systems, which may (partly) contribute to the decrease of DNA concentration. To avoid confusion, we decided to remove Fig. S6c. A reason to include DSB was to maintain the complete pool of DNA variants from one round to the next by enabling outside-of-liposome DNA replication. Note that we omitted DNase I outside of liposomes because it could be taken up inside liposomes during F/T and inhibit in-liposome replication in the next round. In the revised manuscript, page 10, we delete the part “due to rapid extinction of the replicator in the absence of DSB in the tested conditions” and wrote: “This experiment, which we dubbed Con-WT, was conducted in the presence of DSB to maintain the complete pool of replicating DNA variants by enabling outside-of-liposome IVTTR. DNA that replicated in the interior of liposomes but leaked out during F/T (about 50%, see Fig. S6a,b) can potentially be amplified and re-encapsulated in a subsequent round, hence propagate, maintaining the pool of variants that would otherwise be washed out by dilution if DNA replication outside liposomes was disabled (no DSB).”

10. (p10) Fig. 3g was not mentioned in the text, and thus I am not sure what the authors tried to show. However, without any explanation, this panel could cause misinterpretation due to different dilution factors applied to Con- and Bulk-WT. There was generally a higher capacity for replication in Con-WT than in Bulk-WT, and fold-amplification does not necessarily relate to evolution, as opposed to intermittent evolution, where the authors tried to keep the initial DNA concentration constant at each round.

Answer: *We now refer to Fig. 3g in the updated manuscript, on page 10, third paragraph. In New Fig. S8 (page 3 of this letter), we show that DNA can survive serial dilution rounds in Con-WT also when applying 10-fold dilution. We think that the overlay of DNA amplification in the two evolution campaigns is relevant, even if the experimental conditions are not exactly the same.*

11. (p11) The authors wrote, “the abundance of p2 in the bulk evolution experiment decreased faster than the other regions (Fig. 3i),” but it is unclear to my eye. The dynamics of p2 were different clearly from those of ori-L but appeared similar to those of p3 (and ori-R) despite different amplification at the initial round.

Answer: We would like to draw the attention of the reviewer to the rounds 4-6 in panel i), where the concentration of p2 is systematically lower than the other regions (please note the log scale). In Con-WT, the abundance of p2 is slightly lower but the difference is less marked and it does not increase with the number of rounds.

12. (p12) The authors “mapped and extracted the frequency of occurrence of all the point mutations that were detected at a frequency of at least 1% in at least one of the evolutionary rounds (Table 1, Fig. S8).” However, according to the captions of both Table 1 and Fig. S8, they appeared to show mutations detected at frequencies of at least 5%. Which are correct?

Answer: We thank the reviewer for pointing out this unintended discrepancy. Yes, we mapped and extracted the mutations with the frequency of at least 1%, and depicted only the top 5% in Table 1 and Fig. S8, since the complete list was too long to be represented in a main figure. We now realize that inclusion of these extended data in the Supplementary Materials is important. In the modified manuscript, the additional file Supplementary Materials 2 is included and we clarified the related sentence on page 12.

13. (p13) The authors found possible contamination of DNA variants from the Int-WT experiment into the Int-Mut experiment, which may affect the results shown in Fig. 2. I believe that this information should be noted in the “Evolution of self-replicators over multiple rounds of intermittent evolution” section (or p8) to avoid any misinterpretations by readers (especially those who do not read after).

Answer: We agree with the reviewer and revised the manuscript accordingly, on page 8, paragraph 3, as: “Subsequent NGS analysis of Int-Mut suggested that a contamination event from Int-WT(1) evolution affected the course of evolution of Int-Mut (Section Emergence of DNA variants and fixation dynamics).”

14. (p14) In Fig. S10, it is hard to distinguish the 24-bp swapping from other point mutations. Please consider changing the color to highlight the swapping.

Answer: We modified the corresponding figures by highlighting the 24-bp swapping with a different color. We also added a **new Fig. S14** that provides more details about the rearrangements of oriR.

15. (p14) Do the authors have any ideas about how variants of “a large deletion of around 200 bases, preserving the ORF but removing most of the right origin” in the Int-WT experiment could replicate without the replication origin?

Answer: As reported by Gutierrez et al. [1], and by Serrano et al. [2], the complete origins of replication are not absolutely required for TP-primed DNA replication with the Phi29 DNA polymerase. In fact, distant ori-regions become more essential when DNA replication is aided by DSB [2]. Full-length origins can even be replaced by “minimal” origins that consist of a 68-bp region from the left origin of replication [3]. Given that our intermittent evolution campaigns did not involve DSB, these earlier results support our finding that partial removal of the right origin of replication in the int-WT evolutionary campaign could still result in active self-replicators.

[1] Gutiérrez, J., Garmendia, C. & Salas, M. Characterization of the origins of replication of bacteriophage ϕ 29 DNA. *Nucleic Acids Research* 16, 5895–5914 (1988).

[2] Serrano, M., Gutiérrez, J., Prieto, I., Hermoso, J. M. & Salas, M. Signals at the bacteriophage phi 29 DNA replication origins required for protein p6 binding and activity. *EMBO J* 8, 1879–1885 (1989).

[3] Mencía, M., Gella, P., Camacho, A., de Vega, M. & Salas, M. Terminal protein-primed amplification of heterologous DNA with a minimal replication system based on phage Φ 29. *Proceedings of the National Academy of Sciences* 108, 18655–18660 (2011).

16. (p15) The authors suggested using a mutator DNAP variant to accelerate evolution in semi-continuous evolution experiments. Why did not the authors use it in the first place, while using it for intermittent evolution experiments? I will not ask the authors to conduct a new experiment, but if it has already been done (e.g.,

performed but the population went extinct), please disclose the data.

Answer: *We have not performed this experiment. Our finding that DNA self-replication was sustainable for at least 5 rounds in Con-WT (Fig. 4b) was sufficient to support the feasibility of semi-continuous evolution. We suggested using a mutator DNAP variant in a larger context, when high accumulation of mutations is necessary to more rapidly explore the fitness landscape, perhaps under different selection pressures.*

17. (p15, Fig. 4) Please specify what statistical test was used to determine the p-values.

Answer: *This information is provided in the Reporting Summary and has been added in the revised manuscript on page 28:*

“Statistical tests (unpaired t-test, two-tailed P value) shown in Fig. 2c and Fig. 4b were performed using GraphPad Prism.”

18. (p15) The authors stated that their finding of Fig. 4d “suggests a mutational fitness advantage in response to compartmentalization in liposomes,” but I do not think this is necessarily the case. For example, the variant may show a higher initial replication rate as in panel c. Was not the replication already saturated?

Answer: *We thank the reviewer for the insightful question. Indeed, the kinetics of replication may play a role in the performance of the reverse-engineered mutants in bulk and in-liposome IVTTR experiments. We cannot rule out that saturation of replication would explain why the different mutants show similar DNA yield in the bulk experiments but not inside liposomes. However, this can be seen as a manifestation of fitness advantage due to compartmentalization, not as an alternative scenario. Therefore, we think that our statement still holds in the case of differential replication rates between bulk and in-liposome IVTTR.*

19. (p16) The authors investigated the activity of purified polymerases that harbor the enriched mutations in bulk IVTT reactions (Fig. S15) and concluded that “it is unlikely that the amino acid residue substitution in the translated DNAP protein improves its replication activity.” However, because the enhanced IVTTR was so far observed only in in-liposome conditions (Fig. 4), it is conceivable that the purified polymerase variants may act better in liposomes.

Answer: *We agree with the reviewer on the possibility that the purified DNAP variants may act better in liposomes than in bulk. We experimentally tested this hypothesis but could not detect DNA replication, also not with the wild-type polymerase (data not shown). It must be noted that the protocol had to be modified compared to standard in-liposome IVTTR to avoid expression of the p2 gene and, thus competition with the purified polymerase. This result means to us that DNA replication efficiency strongly depends on the precise initial conditions of IVTTR.*

20. (p17) The authors stated, “the discovery that adaptive evolution arose relatively fast (within 10 rounds of evolution)...faster than with the RNA/Q β -replicase system 38,42 (within 100 rounds)...” I think that the “discovery” was referred to as the intermittent evolution experiments. However, in intermittent evolution experiments using an RNA/Q β -replicase system, adaptive evolution could arise just as fast. For example, Mizuuchi et al., ACS Synth. Biol., 2015 (ref 40) demonstrated such evolution within 11 rounds.

Answer: *We thank the reviewer for this kind reminder. Indeed, Mizuuchi et al. demonstrated a fast adaptive evolution of the Q β RNA replicase system, which was already evident within 10 rounds of evolution. This actually proves the authors' point that regeneration of the linear template (RNA or DNA) outperforms circular replication. We concede that our initial claim was not delivered correctly. In the updated manuscript, on p17, we deleted the part “, and faster than with the RNA/Q β -replicase system^{38,42} (within 100 rounds)”. The sentence now reads: “The discovery that adaptive evolution arose relatively fast (within 10 rounds of evolution) compared to repetitive, concatemeric DNA replication⁴⁵, where mutation effects average out due to multiple gene copies per molecule, makes our DNA self-replicating mechanism a good candidate for implementation in an evolving synthetic cell.”*

Subject: Final Revision for Manuscript **NCOMMS-24-26213A**

Title: "Darwinian Evolution of Self-Replicating DNA in a Synthetic Procell"

Author(s): Zhanar Abil, Ana María Restrepo Sierra, Andreea R. Stan, Amélie Châne, Alicia del Prado, Miguel de Vega, Yannick Rondelez & Christophe Danelon

We are pleased to provide our point-by-point response (in *blue* text) to the reviewer's comments (in black text). Modifications in the manuscript or supplementary information are highlighted in *red* text.

REVIEWERS' COMMENTS

Reviewer #2 (Remarks to the Author):

I appreciate the authors' efforts in revising the manuscript, including performing additional experiments, which I believe have significantly improved the research. Most of my previous concerns have been resolved. I now have only a few relatively minor comments, as follows:

1. (Regarding the original comment/answer 2) The new results support evidence of "persistent" DNA replication in liposomes. My only remaining concern is about the different extents of DNA replication observed in the two Bulk-WT experiments (Figs. 3e and S9a). I initially thought that persistent DNA replication may occur in Bulk-WT if a 100-fold dilution is applied because the amplification in the initial round was >100-fold. However, in the new experiment shown in Fig S9, the replication at the initial round was only <5-fold, despite the same initial conditions for both experiments (please correct me if I am wrong). Thus, the decreasing trajectory of DNA concentration and amplification in the new experiment may have resulted from (somehow) significantly different efficiencies of IVTTR. Could the authors provide an explanation for this?

Answer: We have used different batches of PUREfrex during the study and consistent results were obtained. However, in the new experiment shown in Fig. S9, a new batch was employed, which led to an unusually low replication efficiency (<5-fold). We performed a second Bulk-WT (with 100-fold dilution) serial dilution experiment with that same batch, confirming the low amplification (around 10-fold) at the initial round. Subsequent rounds showed higher replication of the p2 gene targeted by qPCR, but this could be attributed to the amplification of a short replicon that emerged in round 1 as shown by gel electrophoresis. These data are presented in the new Fig. S9 (copied below). Despite the low replication efficiency, the appearance of shorter replicons in Bulk-WT further supports the role of compartmentalization in the persistency of full-length DNA in liposomes. We haven't identified yet what causes the drop of IVTTR activity in the new PUREfrex batch and are in contact with the manufacturer on this matter.

We repeated the Bulk-WT (with 100-fold dilution) serial dilution experiment, this time using an older batch of PUREfrex that gives >200-fold replication in bulk IVTTR. Gel analysis of the PCR-amplified DNA shows a clear decreasing trajectory of DNA concentration further supporting our claim that sustained replication of the full-length DNA necessitates liposomes. This time, the takeover by a short replicon is not visible on gel, suggesting that contingency played a role in our evolutionary experiments.

*We modified **Supplementary Note 5** to incorporate these new results. The related claims on page 12 remain valid.*

*"Furthermore, we performed Bulk-WT with 100-fold dilution, as applied in Con-WT (16 h) (**Fig. 3b,c**). In a first experiment, the self-replicator rapidly gets extinct, as observed by qPCR (**Fig. S9a**) and gel electrophoresis (**Fig. S9b**); the amplification efficiency is not sufficient to compensate for the large dilution factor. In a second repeat, a short replicon containing the p2 gene emerged at round 1 and took over mod-ori-p2p3 (**Fig. S9c,d**). Surprisingly, the efficiency of amplification in the initial round*

was unusually low (<10-fold) in both experiments, which was caused by a new batch of PUREfrefx (personal communication with GeneFrontier). We carried out a third repeat, this time with a different (older) batch that showed higher replication efficiency (>100-fold), similar to the other batches of PURE system used throughout this study. A clear decreasing trajectory of mod-ori-p2p3 concentrations was observed on gel (Fig. S9e,f). These results contrast with the persistency of the full-length replicator in Con-WT (16 h) in which the same dilution factor was applied (Fig. 3b), further supporting evidence of the role of liposomes.”

New Fig. S9: Bulk-WT with 100-fold dilution. a,c,e) Trajectories of mod-ori-p2p3 concentrations (left) and amplification folds (right) as measured by qPCR. The p2 gene or both p2 and p3 genes were targeted, as indicated. **b,d,f)** Size analysis of PCR-amplified DNA by agarose gel electrophoresis. The arrowhead indicates the full-length replicator. Results from three repeats are shown. Repeat 1 (a,b), repeat 2 (c,d) and repeat 3 (e,f). A different batch of PUREfrefx with unusually low replication efficiency (<10-fold) was used in repeats 1 and 2.

2. (Regarding the original comment/answer 3) I apologize for my confusing comment about the “faster accumulation” of mutations in Con-WT. I probably meant to write “faster enrichment,” but in any case, now I agree with the authors that this could be explained within the context of Darwinian evolution.

Answer: Thank you.

3. (Regarding the original comment/answer 7) In the new Fig. S1a, the incubation time for the far right lane is missing.

Answer: We added the incubation time.

4. (Regarding the original comment/answer 8) I think that “Figs 2 and 3” in the revised manuscript were meant to be Figs 2 and 4.

Answer: The reviewer is correct, we meant Figs. 2 and 4. We corrected it in the revised manuscript.